# Subthreshold moment analysis of neuronal populations driven by synchronous synaptic inputs

Logan A. Becker[1,2], Francois Baccelli[3,4,5], Thibaud Taillefumier [1,2,3]*

1 Center for Theoretical and Computational Neuroscience, The University of Texas at Austin, Austin, Texas, United States of America, 2 Department of Neuroscience, The University of Texas at Austin, Austin, Texas, United States of America, 3 Department of Mathematics, The University of Texas at Austin, Austin, Texas, United States of America, 4 Département d'informatique, École Normale Supérieure, Paris, France, 5 Institut national de recherche en sciences et technologies du numérique, Paris, France

* ttaillef@austin.utexas.edu

**Data availability statement:** The codes used in this article were deposited in https://github.com/MathNeuro/AONCB_moments.

## Abstract

Even when driven by the same stimulus, neuronal responses are well-known to exhibit a striking level of spiking variability. In-vivo electrophysiological recordings also reveal a surprisingly large degree of variability at the subthreshold level. In prior work, we considered biophysically relevant neuronal models to account for the observed magnitude of membrane voltage fluctuations. We found that accounting for these fluctuations requires weak but nonzero synchrony in the spiking activity, in amount that are consistent with experimentally measured spiking correlations. Here we investigate whether such synchrony can explain additional statistical features of the measured neural activity, including neuronal voltage covariability and voltage skewness. Addressing this question involves conducting a generalized moment analysis of conductance-based neurons in response to input drives modeled as correlated jump processes. Technically, we perform such an analysis using fixed-point techniques from queuing theory that are applicable in the stationary regime of activity. We found that weak but nonzero synchrony can consistently explain the experimentally reported voltage covariance and skewness. This confirms the role of synchrony as a primary driver of cortical variability and supports that physiological neural activity emerges as a population-level phenomenon, especially in the spontaneous regime.

## Author summary

Owing to the sheer complexity of biological networks, identifying the design principles for neural computations will only be possible via the simplifying lens of theory. However, to be accepted as valid explanations, theories need to be implemented in idealized neuronal models that can reproduce key aspects of the measured neural activity.

**Funding:** LB, FB, and TT were supported by the CRCNS program of the National Science Foundation under Award No. DMS-2113213. LB and TT were also supported by the Vision Research program of the National Institutes of Health under Award No. R01EY024071 and by the Division of Mathematical Science of the National Science Foundation under CAREER Award No. NSF-DMS 2239679. The funders had no role in study design, data collection and analysis, decision to publish, or preparation of the manuscript.

**Competing interests:** The authors have declared that no competing interests exist.

Only then can these theories be subjected to experimental validation. In this manuscript, we address this requirement by asking: under which conditions can biophysically relevant neuronal models reproduce physiologically realistic subthreshold activity? We answer this question by focusing on the membrane voltage correlation and skewness, two key statistical signatures of the variable neuronal responses that have been well characterized in behaving mammals. As our core result, we show that the presence of weak but nonzero spiking synchrony is necessary to elicit physiological neuronal responses. The identification of synchrony as a primary driver of neural activity runs counter to the currently prevailing asynchronous state hypothesis, which serves as the basis for many leading neural network theories. Recognizing a central role for synchrony supports that neural computations fundamentally emerge at the collective level rather than as the result of independent parallel processing in neural circuits.

## Introduction

Cortical electrophysiological recordings have revealed that subthreshold neuronal responses exhibit a surprising level of variability in behaving rodents and primates [1]. Even when driven by the same stimulus or when performing the same motor actions, the membrane voltage of neurons in cortical visual or motor areas typically display large, skewed fluctuations [2–5]. We recently argued via mathematical and computational analyses that the observed magnitude of these voltage fluctuations is inconsistent with a purely asynchronous regime of activity [6]. In a purely asynchronous regime, neurons fire independently from one another, so that the probability that a neuron receives synchronous inputs is exceedingly low. To support this argument, we characterize the subthreshold response of a biophysically relevant, conductance-based, neuronal model subjected to synaptic drive with various degrees of synchrony [6]. Given realistic values for synaptic efficacies and input numbers, we identify input synchrony as the main driver of cortical subthreshold variability. This result was derived analytically for a perfect form of synchrony whereby synaptic inputs can activate simultaneously. This result was also confirmed numerically for more realistic forms of synchrony resulting from the waxing and waning of input rates [7,8]. Importantly, the proposed leading role of synchrony in shaping cortical variability is consistent with two further experimental observations: First, the statistical analysis of large-scale *in-vivo* population recordings indicates the reliable presence of weak but non-zero spiking correlations, the statistical signature of synchrony [7,9,10]. Second, *in-vivo* voltage-clamped measurements at the soma reveal large conductance fluctuations that can only be explained by the nearly simultaneous activation of many synaptic inputs [11].

Because of the apparent weakness of the spiking correlations measured *in vivo* [9], the role played by synchrony in cortical variability has typically been overlooked in prevailing modeling approaches [12–14]. However, in keeping with past studies [15,16], our work has shown that when passed through the large number of synaptic inputs, even weak synchrony can be the leading determinant of cortical variability. In this regard, we stress that the level of synchrony required to explain cortical variability are consistent with the amount of spiking correlation reported in [9]. Here, we ask whether input synchrony can consistently and quantitatively explain other features of the subthreshold cortical variability in addition to the observed magnitude of the membrane voltage fluctuations. Specifically, the primary focus of this work will be to show that input synchrony can account for the membrane voltage covariability measured across pairs of jointly recorded neurons. Intracellular recordings of pairs of neurons in both anesthetized and awake animals reveal a high degree of membrane

voltage correlations [17–21]. These simultaneous recordings also reveal that excitatory and inhibitory conductance inputs are highly correlated with each other across neurons and thus, most likely, within the same neuron [19,21]. Aside from explaining these forms of covariability, a secondary focus of our work will be to show that input synchrony is also responsible for the observed skewness of the membrane voltage distribution. *In-vivo* voltage measurements typically exhibit large upward depolarization during spontaneous activity, leading to a baseline level of activity with a positive skewness, which substantially decreases during evoked activity [3,4,22].

To answer our guiding question, we derive exact analytical expressions for the stationary mixed voltage moments of a feedforward pool of neurons driven by synchronous input drives [23,24]. We develop our subthreshold analysis for a variant of classically considered neuronal models, called the all-or-none-conductance-based (AONCB) model, which was introduced in [6]. The hallmark of AONCB neurons is that their synaptic activation mechanism occurs as an all-or-none process rather than as an exponentially relaxing process. The benefit of considering such a mechanism is that it yields equivalent dynamics to those of classical conductance-based models in the limit of instantaneous synapses, while being amenable to exact probabilistic analysis [25,26]. Given a feedforward pool of AONCB neurons, we model their conductance drives as correlated shot noises [23,24]. A benefit of shot-noise-based models compared to classical diffusion-based models is to allow for synchronous synaptic activation events to be temporally separated in distinct impulses [27–29]. Each of these impulses elicits a transient positive conductance fluctuation, whose amplitude is determined by the number and size of the synchronous inputs. We can formalize this picture by modeling conductance drives with varying degree of synchrony as multidimensional jump processes, specifically via compound-Poisson processes [30,31]. In this approach, the degree of input synchrony is entirely captured by the joint distribution of the conductance jumps, which can be quantitatively related to spiking correlations between pairs of inputs.

Our exact analysis, which relies on techniques from queuing theory [32,33], extends our previous results obtained in [6] along two directions: First, it considers synchronous input activity with heterogeneous rates and heterogeneous correlations as opposed to homogeneous populations of exchangeable inputs. Second, it applies to an arbitrary number of feedforward neurons to characterize voltage moments of any order as opposed to being restricted to the mean and variance of the voltage response. Considering biophysically relevant parameters, we leverage these exact results to derive interpretable approximate expressions for voltage correlation between synchronously driven neurons, as well as for their skewness. Utilizing these expressions in combination with simulations, our first main result is to show that weak but nonzero synchrony, in amount that are consistent with physiologically observed spiking correlations, can explain the surprisingly large degree of voltage correlations observed in simultaneous pair recordings. This result is obtained by contrast to pairs of neurons operating in the asynchronous regime, which would require to share an unrealistically large number of inputs. Our second main result is to show that the same amount of input synchrony also explains the large voltage skewness observed during spontaneous activity. These results challenge the prevailing view that neural networks operates in the asynchronous regime and argue for weak but nonzero synchrony to be a primary driver of neural variability, at least in conductance-based neurons. In practice, persistent synchrony may spontaneously emerge in large but finite neural networks, as finite-dimensional interacting dynamics generally exhibit nonzero correlations. However, most theoretical approaches to analyze the impact of these finite-size correlations have been inspired from mean-field techniques, which are derived for infinite-size networks under some Gaussian assumptions [34–40]. It remains unclear if these

approaches can account for the stable emergence of synchrony in large-but-finite networks of conductance-based models.

Technically, to perform our analysis of feedforward AONCB models, we exploit our ability to derive the exact update rule governing shot-noise-driven AONCB dynamics in the limit of instantaneous synapses. Such a limit is obtained by considering that synaptic activation occurs instantly while maintaining the cross-membrane transfer of charge constant. In the limit of instantaneous synapses, the AONCB update rule specifies the evolution of the voltages of a set of $K$ neurons $V_1, \ldots, V_K$ in between two consecutive synaptic input events, as measured at the population level. Because the only source of stochasticity is due to the synchronous shot-noise drive, this evolution is deterministic between the times of these consecutive events, marked be $T_0 < T_1$ for simplicity. In other words, there is a function $\mathcal{F}$ such that

$$\left(V_1(T_1), \ldots V_K(T_1)\right) = \mathcal{F}\left[\left(V_1(T_0), \ldots V_K(T_0)\right), J_0, T_1 - T_0\right],$$

where $J_0$ is the random jump that represents the synaptic event occurring at time $T_0$. Equipped with the above relation, one can derive conservation equations from the time invariance of the stationary shot-noise drive, which states that the distribution of $V_1, \ldots, V_K$ at time $T_0$ and $T_1$ must be identical. In turn, these conservation equations fully characterize the mixed stationary moments of $V_1, \ldots, V_K$ due to the memoryless properties of compound Poisson processes.

As a price for its mathematical tractability, our approach presents key modeling limitations. Specifically, our stationary treatment operates in the limit of instantaneous synapses and, more importantly, assumes an instantaneous form of synchrony, whereby synapses are allowed to activate at the exact same time. Numerical simulations show that when these assumptions are relaxed, our results can still capture the stationary response of synchronously driven AONCB neurons for input models with jittered synchrony [6]. However, it remains unclear whether our results can account for the stationary variability of the neuronal response for more realistic models of synchrony, which exhibit characteristic timescales that vary according to the regime of activity [41–43].

## Methods

### All-or-none-conductance-based neurons

We adopt the all-or-none-conductance-based (AONCB) model introduced in [6] for the subthreshold dynamics of a neuron's membrane voltage. In this model, the membrane voltage of a neuron, denoted by $V$, obeys the first-order stochastic differential equation

$$C\dot{V} = G(V_L - V) + G_e(V_e - V) + G_i(V_i - V) + I, \tag{1}$$

where randomness arises from the stochastically activated excitatory and inhibitory conductances, respectively denoted by $G_e$ and $G_i$ (see Fig 1a). The time-dependent conductances $G_e$ and $G_i$ result from the action of $K_e$ excitatory and $K_i$ inhibitory synapses, respectively: $G_e(t) = \sum_{k=1}^{K_e} G_{e,k}(t)$ and $G_i(t) = \sum_{k=1}^{K_i} G_{i,k}(t)$. In the absence of synaptic input, i.e., when $G_e = G_i = 0$, and for zero external current $I$, the voltage relaxes exponentially toward its leak reversal potential $V_L$ with a time constant $\tau = C/G$, where $C$ denotes the capacitance of the cell membrane and $G$ denotes the passive conductance of the cell [44]. In the presence of synaptic inputs, the membrane voltage fluctuates in response to the transient synaptic currents $I_e = G_e(V_e - V)$ and $I_i = G_i(V_i - V)$, where $V_e$ and $V_i$ denotes the excitatory and inhibitory

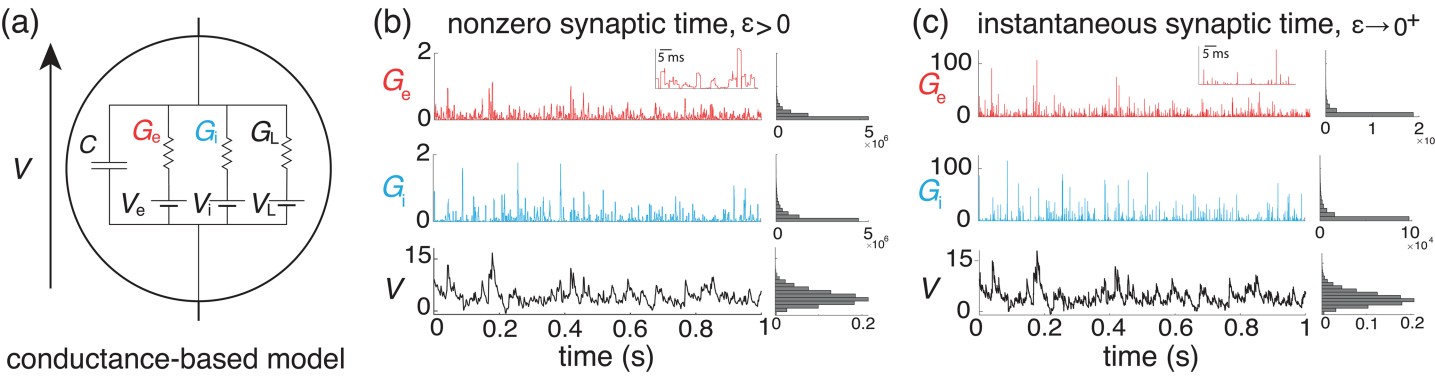

**Fig 1. All-or-none-conductance-based model.** (a) Electrical diagram of conductance-based model for which the neuronal voltage $V$ evolves in response to fluctuations of excitatory and inhibitory conductances $G_e$ and $G_i$. (b-c) The voltage trace and the empirical voltage distribution are only marginally altered by taking the limit $\epsilon \to 0^+$ for short synaptic time constant: $\tau_s = 2$ ms in (b) and $\tau_s = 0.02$ ms in (c). In both (b) and (c), we consider the same compound Poisson process drive with $\rho_e = 0.03$, $\rho_i = 0.06$, and $\rho_{ei} = 0$, and the resulting fluctuating voltage $V$ is simulated via a standard Euler discretization scheme. The corresponding empirical conductance and voltage distributions are shown on the right. The latter voltage distribution asymptotically determines the stationary moments of $V$.

reversal potential, respectively. Without loss of generality, we assume in the following that $V_L = 0$ and that $V_i < V_L = 0 < V_e$.

In the AONCB model, the input spiking activity of the $K_e + K_i = K$ upstream neurons is specified via a $K$-dimensional stochastic point process [30,31]. Let us denote the excitatory components and the inhibitory components of this point process by $\{N_{e,k}(t)\}_{1 \le k \le K_e}$ and $\{N_{i,l}(t)\}_{1 \le l \le K_i}$, respectively, where $l$ and $k$ are upstream neurons' indices. For all upstream neurons $1 \le k \le K_e$ and $1 \le l \le K_i$, $N_{e,k}(t)$ and $N_{i,l}(t)$ are defined as the counting processes that register the spiking occurrences of neurons $k$ and $l$, respectively, up to time $t$. For instance, denoting by $\{T_{e,k,n}\}_{n \in \mathbb{Z}}$ the increasing sequence of the spiking times of excitatory neuron $k$, we have

$$N_{e,k}(t) = \begin{cases} \sum_n \mathbb{1}_{\{0 < T_{e,k,n} \le t\}} & \text{if} \quad t > 0, \\ -\sum_n \mathbb{1}_{\{t < T_{e,k,n} \le 0\}} & \text{if} \quad t \le 0, \end{cases}$$

where $\mathbb{1}_A$ denotes the indicator function of set $A$ ($\mathbb{1}_A(x) = 1$ if $x$ is in $A$ and $\mathbb{1}_A(x) = 0$ if $x$ is not in $A$). Note that by convention, we label spikes so that $T_{e,k,0} \le 0 < T_{e,k,1}$. Similar definitions hold for inhibitory synapses in terms of the counting processes $N_{i,l}(t)$, $1 \le l \le K_i$. The hallmark of AONCB models is that their synaptic conductances operate all-or-none with a common activation time $\tau_s$. Given the point process $N_{e,k}(t)$, this amounts to considering that the conductance process $G_{e,k}(t)$ follows

$$G_{e,k}(t) = g_{e,k}\big(N_{e,k}(t) - N_{e,k}(t - \tau_s)\big), \tag{2}$$

where $g_{e,k} \ge 0$ is the synaptic conductance of the excitatory input $k$. The above equation prescribes that at each spike delivery to synapse $k$, the conductance $G_{e,k}$ instantaneously increases by an amount $g_{e,k}$ for a period $\tau_s$, after which it decreases by the same amount (see Fig 1b). Thus, the synaptic response prescribed by Eq (2) is all-or-none as opposed to being graded. This all-or-none behavior was introduced in [6] because it allows one to derive integral expressions for the stationary mean and variance of the voltage, even when the neuron is driven by synchronous synaptic inputs. Again, similar definitions hold for inhibitory synapses

in terms of the counting processes $N_{i,l}(t)$ and the inhibitory synaptic conductances $g_{i,l}$, $1 \leq l \leq K_i$.

## Jump-process model for synchronous synaptic inputs

To model input synchrony, we consider that distinct synapses can activate (and therefore deactivate) at exactly the same time. In other words, the counting processes associated to distinct synaptic inputs, say $N_{e,k}(t)$ and $N_{i,l}(t)$, are allowed to share points, meaning that it may be that $T_{e,k,m} = T_{i,l,n}$ for some spike indices $m$ and $n$. Because synaptic activations can be simultaneous, it is convenient to distinguish between synaptic event times and synaptic-event sizes. Synaptic-event times mark all these instants when at least one synaptic input activates; whereas synaptic-event sizes capture the total conductance increase at a synaptic event. Accordingly, we define the increasing sequence of synaptic event times $\{T_n\}_{n \in \mathbb{Z}}$ by temporally ordering the set of synaptic spiking times

$$\{T_{e,k,m}, T_{i,l,n} \mid m \in \mathbb{Z}, n \in \mathbb{Z}, 1 \leq k \leq K_e, 1 \leq l \leq K_i\},$$

without multiple counts and so that by convention $T_0 \leq 0 < T_1$. Denoting the counting process that registers synaptic events by $N(t)$, observe that in general, we have $N(t) \leq \sum_k N_{e,k}(t) + \sum_l N_{i,l}(t)$. This inequality becomes strict whenever the process $N(t)$ counts a single event when many synapses activate at the same time, say $T_n$. To specify the synaptic-event size at time $T_n$, let us define $\{0, 1\}$-valued binary variables $X_{e,k,n}$ and $X_{i,l,n}$ such that $X_{e,k,n} = 1$ if and only if excitatory synapse $k$ activates at time $T_n$ and $X_{i,l,n} = 1$ if and only if inhibitory synapse $l$ activates at time $T_n$. The synaptic-event size at time $T_n$ is then defined as the two-dimensional jump

$$(G_{e,n}, G_{i,n}) = \left( \sum_{k=1}^{K_e} g_{e,k} X_{e,k,n}, \sum_{l=1}^{K_i} g_{i,l} X_{i,l,n} \right). \tag{3}$$

With these notations, one can then write the time-dependent conductance that drive an AONCB neuron as a jump process

$$\left( G_e(t), G_i(t) \right) = \left( \sum_{n=N(t-\epsilon\tau)+1}^{N(t)} G_{e,n}, \sum_{n=N(t-\epsilon\tau)+1}^{N(t)} G_{i,n} \right),$$

where $N(t)$ is the point process governing the synaptic-event times $T_n$ and where the jumps $(G_{e,n}, G_{i,n})$ specify the corresponding synaptic-event sizes.

To fully define our jump-process-based model for synchrony, it only remains to specify the behaviors of $N(t)$ and $(G_e, G_i)_{n \in \mathbb{Z}}$ as random processes. Here, as in [6], we make the simplifying assumptions that: (*i*) $N(t)$ is a Poisson process with constant event rate $b$ and (*ii*) that the $K$-dimensional vectors of synaptic activation variables $(\{X_{e,k,n}\}_{1 \leq k \leq K_e}, \{X_{i,l,n}\}_{1 \leq l \leq K_i})_{n \in \mathbb{Z}}$ are independently and identically distributed. Note that assumption (*ii*) implies that the conductance jumps $(G_{e,n}, G_{i,n})_{n \in \mathbb{Z}}$ are independently and identically distributed on the positive orthant $\mathbb{R}^+ \times \mathbb{R}^+$, with some joint distribution denoted by $p_{G_e, G_i}$. These assumptions correspond to neglecting any form of temporal dependencies in the inputs. Although this neglect restricts our modeling to an unrealistically precise form of synchrony, we justified in [6] that this approach is predictive of the response of AONCB neurons to more realistic, jittered, synchronous inputs. Observe that the above approach generalizes the framework proposed in [6]

as we consider arbitrary distribution of $(\{X_{e,k}\}_{1 \le k \le K_e}, \{X_{i,l}\}_{1 \le l \le K_i})$ over $\{0,1\}^{K_e} \times \{0,1\}^{K_i}$. In particular, we do not assume that the inputs are exchangeable.

### Input synchrony and spiking correlations

In our jump-process-based framework, input synchrony follows from the simultaneous activation of inputs at the exact same time. This notion of synchrony, which is defined in continuous time, can be related to the more familiar notion of spiking correlations, which is a measure of synchrony between inputs in discrete time. Specifically, we show in [6] that within our jump-process-based framework, the spiking correlation between any two inputs $k$ and $l$ is defined as

$$\rho_{\alpha\beta,kl} = \frac{\mathbb{E}\left[X_{\alpha,k}X_{\beta,l}\right]}{\sqrt{\mathbb{E}\left[X_{\alpha,k}\right]\mathbb{E}\left[X_{\beta,l}\right]}} \quad \text{with} \quad \alpha,\beta \in \{e,i\} \tag{4}$$

where $\mathbb{E}\left[\cdot\right]$ denotes the expectation with respect to the distribution of synaptic activation variables $(\{X_{e,k}\}_{1 \le k \le K_e}, \{X_{i,l}\}_{1 \le l \le K_i})$. Note that we may omit referencing whether the inputs are excitatory or inhibitory for notational simplicity, as will be the case in the following.

The definition of spiking correlations given in Eq (4) allows one to establish a direct link between spiking correlation and input synchrony within our jump-process-based framework (see Fig 2). When synapses operate asynchronously, distinct inputs $k$ and $l$ activate in isolation, so that $X_k X_l = 0$ with probability one over synaptic events, which implies $\rho_{kl} = 0$. By contrast, in the presence of synchrony, inputs $k$ and $l$ coactivate reliably, so that $X_k = X_l = 1$ with nonzero probability over synaptic events, which implies $\rho_{kl} > 0$. In the extreme case of

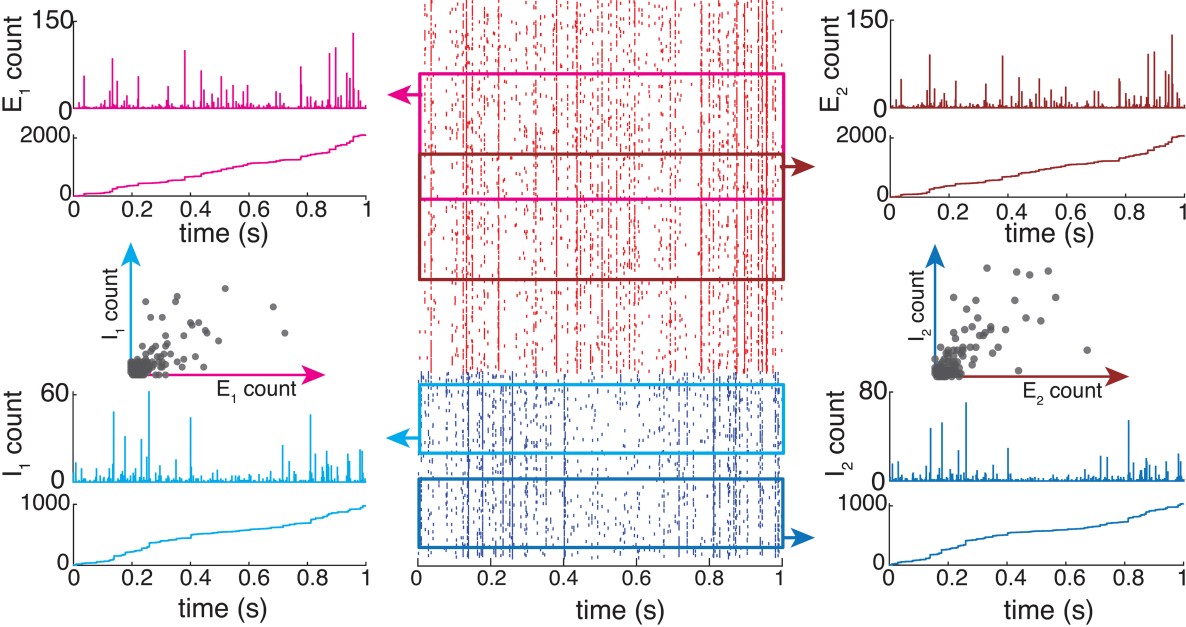

**Fig 2. Spiking correlations represented by a jump-process-based framework**. Excitatory (red) and inhibitory (blue) spikes are drawn from a jump-process with $p_{kl} > 0$, and a total conductance $G_{e/i}$ given by their weighted sums. Inputs for the two example neurons are correlated (or shared) within and across both the excitatory and inhibitory populations, and between both neurons, as expressed through their jump-count distributions.

full synchrony, $X_k = X_l = 1$ with probability one over synaptic events and $\rho_{kl} = 1$. Incidentally, this reveals that our jump-process-based framework can only account for positive spiking correlations, a key limitation of our approach. Note that, as the activation variables $X_k$ are $\{0, 1\}$-valued, higher-order correlation coefficients can also be defined via

$$\rho_{k_1 \dots k_n} = \frac{\mathbb{E}\left[X_{k_1} \dots X_{k_n}\right]}{\sqrt[n]{\mathbb{E}\left[X_{k_1}\right] \dots \mathbb{E}\left[X_{k_n}\right]}}. \tag{5}$$

For all $n>0$, the above coefficient satisfies $0 \le \rho_{k_1 \dots k_n} \le 1$ and is nonzero as soon as all $n$ inputs coactivate reliably across synaptic events. Fully specifying the distribution of the vector of activation variables $(X_1, \dots, X_K)$ amounts to know all the activation probability $p_k = \mathbb{E}\left[X_k\right]$ and all the correlation coefficients $\rho_{k_1 \dots k_n}$, $1 \le k_1, \dots, k_n \le K$, up to order $K$.

Within our modeling framework, synchrony only impacts the dynamics of AONCB neurons via the conductance jumps $(G_e, G_i)$, which are linear combinations of the activation variables $\{X_{e,k}\}_{1 \le k \le K_e}$ and $\{X_{i,l}\}_{1 \le l \le K_i}$ by virtue of Eq (3). Thus, assuming all the synaptic input conductances $\{g_{e,k}\}_{1 \le k \le K_e}$ and $\{g_{i,l}\}_{1 \le l \le K_i}$ are known, one can parametrize the jump distribution $p_{G_e,G_i}$ in terms of all the spiking correlation coefficients up to order $K$. However, such a parametrization is excessively cumbersome, and for the purpose of obtaining exact results, we will always consider $p_{G_e,G_i}$ as a known arbitrary quantity. In turn, we will see that under the biophysically relevant assumptions of small input weights, approximate results can be directly stated in terms of the spiking correlation coefficients.

## Marcus dynamics in the limit of instantaneous synapses

We obtain our exact results about the variability of synchronously driven AONCB neurons in the limit of instantaneous synapses. Informally, this corresponds to considering that synaptic inputs act instantly to transfer charges within a neuron. In order to define this limit regime formally, let us introduce the dimensionless synaptic weights $w_{e,k} = g_{e,k}\tau_s/C$ and $w_{i,k} = g_{i,k}\tau_s/C$ and consider the associated dimensionless conductance jumps $W_{e,n} = G_{e,n}\tau_s/C$ and $W_{i,n} = G_{i,n}\tau_s/C$, $n \in \mathbb{Z}$. Denoting by $\epsilon = \tau_s/\tau > 0$ the ratio of the duration of synaptic activation relative to the passive membrane time constant, one can write the conductance jump process in terms of the dimensionless jumps $(W_{e,n}, W_{i,n})_{n \in \mathbb{Z}}$ as

$$\left(G_e(t), G_i(t)\right) = \frac{G}{\epsilon}\left(\sum_{n=N(t-\epsilon\tau)+1}^{N(t)} W_{e,n}, \sum_{n=N(t-\epsilon\tau)+1}^{N(t)} W_{i,n}\right), \tag{6}$$

thereby exhibiting a natural scaling with respect to the parameter $\epsilon$. The limit of instantaneous synapses corresponds to taking $\epsilon \to 0^+$ while holding the dimensionless synaptic weights constant (see Fig 1c). Such a scaling maintains the charge transfer during a synaptic event, thereby preserving the impact of synaptic activations on AONCB dynamics as $\tau_s = \epsilon\tau \to 0^+$. In the following, we denote the common distribution of the independent variables $(W_{e,n}, W_{i,n})_{n \in \mathbb{Z}}$ by $p_{W_e,W_i}$ and just as for $p_{G_e,G_i}$, we consider the latter distribution as a known quantity in our calculations.

Assuming that the jump distribution $p_{W_e,W_i}$ is known, one can exploit the $\epsilon$-scaling in Eq 6 to develop a simplified analytical treatment of the AONCB neuron dynamics in response to synchronous inputs. This simplified treatment follows from the fact that when $\epsilon \to 0^+$, the driving conductance process becomes a two-dimensional shot noise [6], i.e., the temporal derivative of a two-dimensional compound Poisson processes $Z(t) = \left(Z_e(t), Z_i(t)\right)$ [30,31].

Specifically, we have

$$\lim_{\epsilon \to 0^+} \left( \frac{G_e(t)}{G}, \frac{G_i(t)}{G} \right) = \frac{d}{dt} \big( Z_e(t), Z_i(t) \big), \quad \big( Z_e(t), Z_i(t) \big) = \left( \sum_n^{N(t)} W_{e,n}, \sum_n^{N(t)} W_{i,n} \right),$$

where the notion of convergence can be made precise but is irrelevant for practical purposes. Shot-noise-driven models are amenable to exact analysis, albeit with some caveats as one can generally define several notions of solution [27,45]. To identify the physical solution, one considers dynamics subjected to a regularized version of the shot noise, whose regularity is controlled by a positive parameter, say, $\alpha > 0$ [25,26]. Then, shot-noise-driven dynamics are recovered in the limit of vanishing regularization, typically in the asymptotic regime $\alpha \to 0^+$. Our previously introduced parameter $\epsilon = \tau_s/\tau$ precisely plays the role of such a regularizing parameter. Correspondingly, one can derive the shot-noise-driven dynamics of AONCB neurons by considering solutions to Eq (1) for $\epsilon > 0$ and then taking $\epsilon \to 0^+$.

The above approach yields the so-called Marcus dynamics for AONCB neurons, which can be stated concisely for constant current $I$ as follows: The membrane voltage $V$ relaxes exponentially toward the resting potential $I/G$ with time constant $\tau$, except when subjected to synaptic impulses at times $\{T_n\}_{n\in\mathbb{Z}}$. At these times, the voltage $V$ updates discontinuously according to $V(T_n) = V(T_n^-) + J_n$, where the jumps are given via the Marcus rule:

$$J_n = \left( \frac{W_{e,n} V_e + W_{i,n} V_i}{W_{e,n} + W_{i,n}} - V(T_n^-) \right) \left( 1 - e^{-(W_{e,n} + W_{i,n})} \right). \tag{7}$$

Observe that the above rule implies that the voltage must lie within $(V_i, V_e)$, the allowed range of variation for $V$. Note that such a formulation also specifies an exact even-driven simulation scheme given knowledge of the synaptic activation times and sizes $\{T_n, W_{e,n}, W_{i,n}\}_{n\in\mathbb{Z}}$ [46]. We adopt the above Marcus-type numerical scheme, which differs from classical Euler-type discretization scheme, in all the simulations that involve instantaneous synapses. More generally, the above Marcus formulation of AONCB dynamics, which combines exponential relaxation and random jump discontinuities, is at the root of all the exact results that we derive in the following.

## Feedforward population models

It is mostly a matter of notations to generalize the Marcus jump dynamics given in Eq (7) to a population of feedforward neurons. To see this, let us consider a set of neurons $A$ with cardinality denoted by $|A|$. Each neuron $a \in A$ may receive inputs from a set of $K_e$ excitatory inputs and $K_i$ inhibitory inputs. Accordingly, each neuron $a \in A$ experiences jumps whose sizes depend on the input activation variables $\{X_{e,k}\}_{1 \le k \le K_e}$ and $\{X_{i,l}\}_{1 \le l \le K_i}$ via their own dimensionless excitatory and inhibitory weights $w_{e,a,k}$ and $w_{i,a,k}$. This corresponds to considering $|A|$ shot-noise-driven AONCB neuron dynamics with $(2|A|)$-dimensional jumps

$$\left( W_{e,a}, W_{i,a} \right)_{a\in A} = \left( \sum_{k=1}^{K_e} w_{e,a,k} X_{e,k}, \sum_{l=1}^{K_i} w_{i,a,l} X_{i,l} \right)_{a\in A}.$$

Thus, specifying a shot-noise population model requires assuming knowledge of a population-level $(2|A|)$-dimensional distribution of the jumps $\left( W_{e,a}, W_{i,a} \right)_{a\in A}$, denoted by $p_{W_{e,A}, W_{i,A}}$, instead of a two-dimensional distribution $p_{W_e, W_i}$.

With the above notations, one can generalize Marcus dynamics to neural feedforward populations, which do not connect recurrently to one another (see Fig 3). Such a population of neurons still collectively experiences synaptic events at time $\{T_n\}_{n \in \mathbb{Z}}$, which follow a Poisson process with overall rate $b$. Although the rate $b$ depends on the set of neurons $A$, we will not refer to this dependence explicitly unless required. The key property to observe is that given two sets of neurons $A$ and $B$ such that $A \subset B$, we have

$$b_A = b_B \mathbb{E}\left[\mathbb{1}_{\{\sum_{a \in A} W_{e,a} + W_{i,a} > 0\}}\right] \leq b_B, \tag{8}$$

where the expectation is with respect to the jumps $\left(W_{e,a}, W_{i,a}\right)_{a \in B}$. This simply means that the rate associated to the smaller population is obtained by subsampling the rate of the larger population. Incidentally, observe that one can recover the individual input rate, which we denote $r_{\alpha,k}$ for the $k$-th synapse of type $\alpha \in \{e, i\}$, via:

$$r_{\alpha,k} = b \mathbb{E}\left[\mathbb{1}_{\{X_{\alpha,k} > 0\}}\right]. \tag{9}$$

With these remarks in mind, note that in between two consecutive synaptic events, each neuron $a \in A$ relaxes independently toward its resting potential $I_a/G_a$, with membrane time constant $\tau_a = C_a/G_a$. At synaptic-event times $T_n$, the population voltages update discontinuously as

$$\left(V_a(T_n)\right)_{a \in A} = \left(V_a(T_n^-) + J_{a,n}\right)_{a \in A}, \tag{10}$$

where for all $a \in A$, the individual jumps $J_{a,n}$ are still given via the Marcus update rule given in Eq (7). Thus, the only additional complication to the single-neuron case follows from the multidimensionality of the collective jump update in (10).

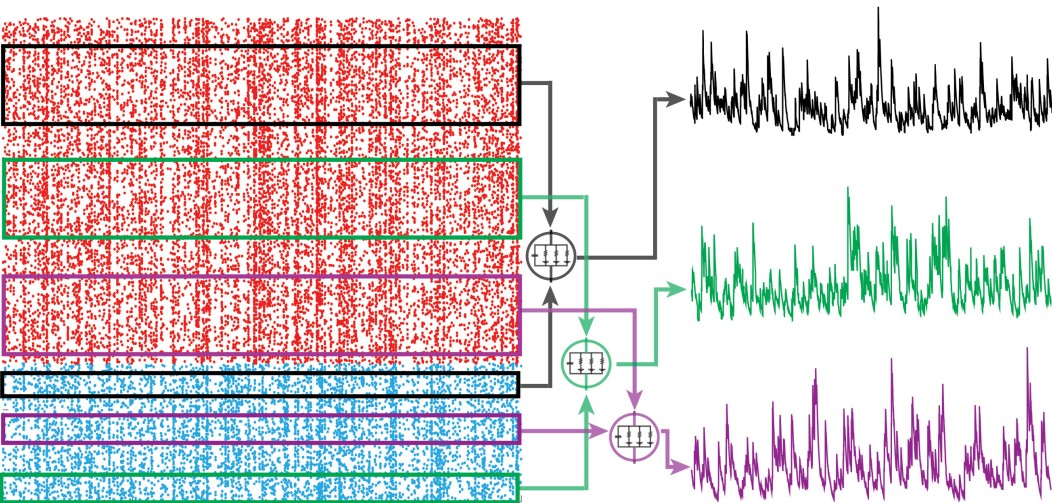

**Fig 3. Feedforward population model:** AONCB neurons receive $K_e$ excitatory (red) and $K_i$ inhibitory (blue) feedforward inputs from a near-infinite pool of possible inputs $K > K_e + K_i \gg 0$. Neurons one (black), two (green), and three (purple) are driven by different, but correlated, spiking inputs resulting in correlated subthreshold activity.

In the following, we consider the stationary version of the process governed by (10). Intuitively, one obtains the stationary version of a process by assuming that the initial condition has been pushed infinitely far in the past. As a result, the initial condition has no bearing on the value of the process at, say, time $t = 0$, which can be viewed as a "typical" time for an otherwise time-shift invariant process. The latter property of time-shift invariance formally defines stationary processes, which can be analyzed via a variety of mathematical results. In this work, we utilize one such result, the so-called PASTA principle from queuing theory, which stands for "Poisson arrivals see time averages" [32,33]. Within the context of AONCB neurons with instantaneous synapses, these arrivals refer to synaptic input activations, assumed to follow a Poisson process.

## PASTA (<u>P</u>oisson <u>A</u>rrivals <u>S</u>ee <u>T</u>ime <u>A</u>verages) principle

In a nutshell, the PASTA principle states that sampling a stationary process $V(t) = (V_a(t))_{a \in A}$ driven by Poisson input arrivals at a typical time, say $V(0)$ at $t = 0$, is equivalent to sampling the same process just before an input arrival, say $V(T_1^-)$, where $T_1$ denotes the first input arrival time after $t = 0$. Making this point rigorous requires the introduction of the concept of Palm distribution, which considers stationary point processes at a typical point, i.e., at an input spike, rather than at a typical time, i.e., in between input spikes [32]. A defining property of Poisson processes is that their Palm distribution is the same as their stationary distribution. As a result, stationary expectations, i.e., expectations at a typical time, can be evaluated as expectations at a typical point, justifying the PASTA principle (see Fig 4).

The PASTA principle is valid for all processes $V$ that are driven by compound Poisson processes $Z$, including those with multidimensional jumps [33]. Here, "driven" means that the future history of arrivals $\{Z(s) - Z(t)\}_{s \geq t}$ is independent from the past history of the process $\{V(s)\}_{s < t}$, whereas the future history of the process $\{V(s) - V(t)\}_{s \geq t}$ generally depends on the past history of arrivals $\{Z(s)\}_{s < t}$. In particular, and most importantly for this work, the PASTA principle applies to shot-noise-driven AONCB neurons. As a result, one can evaluate the stationary moments of membrane voltages as expectations with respect to the associated Palm distribution. Specifically, denoting by $A_n = \{a_1, \dots, a_n\}$ a multiset of elements in $A$, which allows for multiple instances of its elements (i.e., for repeated neuronal indices in our case), let us define the corresponding $n$-th order, shifted, stationary moment as

$$\mu_{A_n} = \mathbb{E}\left[\prod_{a \in A_n}(V_a(0) - I_a/G_a)\right].$$

Then, the PASTA principle implies that we must have

$$\mu_{A_n} = \mathbb{E}\left[\prod_{a \in A_n}(V_a(T_1^-) - I_a/G_a)\right] = \mathbb{E}\left[\prod_{a \in A_n}(V_a(T_0^-) - I_a/G_a)\right], \tag{11}$$

where $T_0$ and $T_1$, denote the two consecutive synaptic-event times framing $t = 0$: $T_0 < 0 < T_1$. The main point of considering two consecutive times $T_0$ and $T_1$ is that: (*i*) $V(T_1^-)$ follows from $V(T_0^-)$ via application of a single Marcus update rule and (*ii*) the inter-event interval $S_1 = T_1 - T_0$ is an exponentially distributed variable that is independent of $V(T_0^-)$. More precisely, given $V(T_0^-)$, we have

$$\prod_{a \in A_n}(V_a(T_1^-) - I_a/G_a) = \prod_{a \in A_n}\left((J_{a,0} + V_a(T_0^-) - I_a/G_a)e^{-\frac{S_1}{\tau}}\right), \tag{12}$$

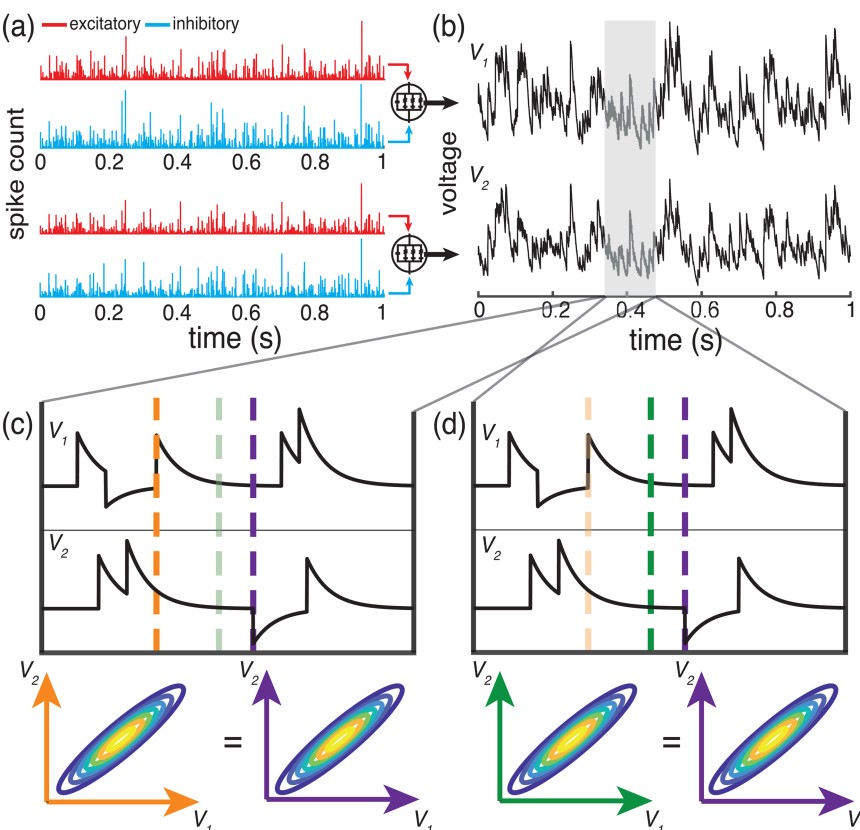

**Fig 4. Sampling the voltage of an AONCB neuron at a typical time $T$ is equivalent to sampling the same process just before an input arrival.** (a) Example distinct but correlated excitatory (red) and inhibitory (blue) inputs for 2 neurons with activity $V_1$ and $V_2$ (b). (c-d) Zoomed in example voltage activity for both neurons with corresponding joint voltage distributions sampled at dashed lines. Joint voltage densities are equivalent whether sampling just before an input arrival (c; orange and purple lines) or at some arbitrary time $T$ (d; green and purple lines).

where $J_{a,0}$ is a random jump specified by Eq (7) for neuron $a$ and where $S_1 = T_1 - T_0$ is an exponential waiting time with mean $1/b$. Then, one can leverage Eq (12) in combination with Eq (11) to find fixed-point equations for the shifted moments $\mu_{A_n}$. It turns out that these equations define an exactly solvable triangular system of equations for the shifted moments $\mu_{A_n}$. We derive and solve such a system in S1 Text, S1 Appendix, ultimately yielding generic expressions for the centered moments $M_{A_n} = \mathbb{E}\left[\left(V_{a_1}(0) - m_{a_1}\right) \dots \left(V_{a_n}(0) - m_{a_n}\right)\right]$, where $m_a = \mu_a + I_a/G_a$ denotes the stationary mean of the voltage $V_a$.

## Biophysical parameters

To investigate the biophysical relevance of our analysis, we use realistic estimates for the various parameters featured in the AONCB neurons. Specifically, we assume the common values $\tau = 15\text{ms}$ for the passive membrane time constant and $V_i = -10\text{mV} < V_L = 0 < V_e = 60\text{mV}$ for reversal potentials. Given these common assumptions, determining the dynamics of a synchronously driven AONCB neuron still requires to specify realistic values for the dimensionless synaptic weights $w_{\alpha,k}$ and for the spiking correlation coefficients $\rho_{\alpha\beta,kl}$ across synaptic inputs.

To estimate synaptic weights, we consider that when delivered to a neuron close to its resting state, unitary excitatory inputs cause peak membrane fluctuations of up to $\simeq 0.5$mV at the soma, attained after a peak time of $\simeq 5$ms. Such fluctuations correspond to typically large *in-vivo* synaptic activations of thalamo-cortical projections in rats [47]. Although activations of similar amplitudes have been reported for cortico-cortical connections [48,49], recent large-scale *in vivo* studies have revealed that cortico-cortical excitatory connections are typically much weaker [50,51]. At the same time, these studies have shown that inhibitory synaptic conductances are about fourfold larger than excitatory ones, but with similar timescales. Fitting these values within the framework of AONCB neurons for $\epsilon = \tau_s/\tau \simeq 1/4$ reveals that the largest possible synaptic inputs correspond to dimensionless weights $w_e \simeq 0.01$ and $w_i \simeq 0.04$. Following on [50,51], we consider that the comparatively moderate cortico-cortical recurrent connections are an order of magnitude weaker than typical thalamo-cortical projections, i.e., $w_e \simeq 0.001$ and $w_i \simeq 0.004$. Such a range is in keeping with estimates used in [52,53].

With respect to input synchrony, physiological estimates of the spiking correlations are typically reported to be weak, with coefficients ranging from 0.01 to 0.04 [9,14,42]. It is important to note that such weak values do not warrant the neglect of correlations as the number of synaptic connections can be large in cortex. For instance, as we will see for AONCB neurons, synchrony significantly impacts voltage variability as soon as $(K_e - 1)\rho_e$ is larger than or about 1. Assuming the lower estimate of $\rho_e \simeq 0.01$, this criterion is achieved for $K_e \simeq 100$ inputs, which is well below the typical number of excitatory synapses for cortical neurons ($K_e \simeq 10^4$). Although the relevance of weak correlations to neural variability has been noted before [15,16]. there can be competing explanatory factors. In the following, we argue via biophysical considerations that the leading competing explanation, synaptic heterogeneity, can only account for the small fraction of the observed subthreshold variability.

By considering typical values for synaptic weights, we overlook the role of synaptic heterogeneity in shaping neuronal covariability [54,55]. This is an important caveat as for any synchronous input model, one can design an equivalent asynchronous input model but with heterogeneous synapses [6]. As we will later justify in the discussion, in asynchronous models, the impact of synaptic heterogeneity on neuronal variability is mediated by the squared coefficient of variation of the synaptic weights: $\mathrm{CV}^2[w] = \mathbb{V}[w_e]/\mathbb{E}[w_e]^2$. Experimental measurements of $\mathrm{CV}[w]$ have consistently been reported to be below one [56,57], implying that synaptic heterogeneity can induce up to a two-fold increase in the observed subthreshold variability compared to asynchronous, homogeneous models. Our previous work [6] showed that this is inconsistent with the high level of observed subthreshold variability in cortical neurons with moderate synaptic weights, which would require $\mathrm{CV}^2[w]$ to be at least an order of magnitude larger. This supports that synchrony is the main determinant of neuronal variability. For this reason, although we will derive all our results assuming synaptic heterogeneity, we will primarily discuss their practical consequences assuming typical moderate weights.

## Numerical simulation methods

Several techniques have been proposed to model and simulate synchronous spiking activity besides our jump-process-based modeling framework. For instance, some of these techniques generate synchrony by leveraging the superposition and thinning properties of Poisson processes [58], by considering Poisson processes with multidimensional stochastic intensities [59], or by considering doubly-stochastic Cox processes [60,61]. Although our modeling framework relies on compound Poisson processes, its practical numerical implementation can be understood within the latter doubly-stochastic approach. In this approach, synchrony

arises from governing the spiking mechanisms of distinct neurons with correlated rate processes. Given a $L \times L$ block-structured synaptic-input correlation matrix $\{\rho_{ij}\}$, $1 \le i,j \le L$, each homogeneous subpopulation (block) of inputs is assigned a given rate process $t \mapsto \Theta_i(t)$, $1 \le i \le L$. For compound Poisson processes, these rate processes are actually formally defined as completely random measures, which can be interpreted as directing, infinite-dimensional, de Finetti processes [6].

In practice, however, we only simulate the rate processes $\Theta_i$, $1 \le i \le L$, with finite temporal resolution in bins of durations $\Delta t = 10^{-4}$s. As a result, the random measure associated to each population is replaced be a bin-indexed i.i.d. process: $\theta_{i,j} = \Theta_i(((j-1)\Delta t, j\Delta t])$, where $j$ is the bin index. In the following, we drop the dependence on the time index as the vectors $\{\theta_{1,j}, \ldots, \theta_{L,j}\}_i$ are i.i.d. over time. Then the crux of the method is to sample vectors $\{\theta_1, \ldots, \theta_L\}$ so that these can be used as mixing variables to generate spiking counts $\{k_1, \ldots, k_L\}$ with the desired synchrony structure. Note that at finite resolution, these spike counts are obtained via mixing of the binomial approximations for Poisson processes. This means that the probability law of the counts $\{k_1, \ldots, k_L\}$ is given by

$$p_{k_1, \ldots, k_L} = \mathbb{E}\left[\prod_{i=1}^{L} \binom{K_i}{k_i} \theta_i^{k_i}(1-\theta_i)^{K_i-k_i}\right], \tag{13}$$

where the expectation is with respect to the law of the mixing variables $\{\theta_1, \ldots, \theta_L\}$.

We specify the law of $\{\theta_1, \ldots, \theta_L\}$ via Gaussian copulas, for which many sampling methods are available. Specifically, we consider the $L$-dimensional Gaussian copula defined as

$$C_\Sigma(u_1, \ldots, u_L) = \Phi_\Sigma(\Phi_1^{-1}(u_1), \ldots, \Phi_1^{-1}(u_L)),$$

where $\Phi_\Sigma$ denotes the $L$-dimensional cumulative distribution of the centered Gaussian with correlation matrix $\Sigma$. The matrix $\Sigma$ will be later chosen so that given a choice of marginals for the mixing variables $\theta_i$, $1 \le i \le L$, the desired correlation structure is numerically achieved. Given that we consider the binomial mixing model (13), we choose the cumulative marginal distribution $B_j$ of $\theta_j$ to follow a beta distribution $B_i \sim \mathrm{Be}(\alpha_i, \beta_i)$ with parameters

$$\alpha_i = r_i \Delta t \left(\frac{1}{\rho_{ii}} - 1\right) \quad \text{and} \quad \beta_i = (1 - r_i \Delta t)\left(\frac{1}{\rho_{ii}} - 1\right), \quad 1 \le i \le L.$$

As shown in [6], such parameter choices ensure that the mean spike count is exactly $r_i \Delta t$ in each bin, whereas the subpopulation spiking correlation is $\rho_{ii} + o(\Delta t)$ in each bin, so that the model becomes exact in the limit $\Delta t \to 0^+$. Then, we sample $\{\theta_1, \ldots, \theta_L\}$ according to the copula cumulative distribution

$$C(\theta_1, \ldots, \theta_L) = C_\Sigma(B_1(\theta_1), \ldots, B_L(\theta_L)),$$

where the covariance matrix $\Sigma$ has unit diagonal and off-diagonal terms chosen so that the resulting distribution satisfies

$$\mathbb{E}\left[\theta_i \theta_j\right] = \sqrt{r_i r_j} \rho_{ij} \Delta t, \quad 1 \le i \ne j \le L.$$

This process is fast as we numerically found that given fixed noncorrelation parameters, for all $1 \le i \ne j \le L$, the coefficients $\sigma_{ij}$ each appear to be related to $\log \rho_{ij}$ via the same affine function.

We use the synchronous spike trains generated via the above copula method to drive AONCB neurons. The resulting simulated traces are then used to empirically estimate the various stationary moments considered in this work. Empirically estimating the stationary moments of a population of neurons $A$ only requires simulating voltages on the synaptic event times $T_n$, $n \geq 0$, when at least one neuron from $A$ receives a synaptic input. These times are given as $T_n = j_n \Delta t$ where we have defined

$$j_n = \inf \left\{ j > j_{n-1} \,\bigg|\, \sum_{i=1}^{L} k_{i,j} \geq 1 \right\} \quad \text{with} \quad j_1 = \inf \left\{ j \geq 0 \,\bigg|\, \sum_{i=1}^{L} k_{i,j} \geq 1 \right\},$$

Remember that in the above definitions, $k_{i,j}$ denotes the spike count of the subpopulation $i$ in time bin $j$. At the cost of defining a larger number of subpopulations $L$, we can always define the $L$ homogeneous input subpopulations so that they each correspond to a single typical synaptic weight $w_{i,a}$ for all $i$, $1 \leq i \leq L$ and all $a \in A$. Then, the Marcus update rule (10) applies at $T_n$, $n \geq 0$, with jumps $J_{a,n}$, $a \in A$, computed for the dimensionless conductance jumps

$$\left( W_{e,a,n}, W_{i,a,n} \right) = \left( \sum_{i=1}^{L} k_{i,j_n} \mathbb{1}_{\{w_{i,a} > 0\}} w_{i,a}, \sum_{i=1}^{L} k_{i,j_n} \mathbb{1}_{\{w_{i,a} < 0\}} w_{i,a} \right), \quad a \in A.$$

In turn, the sequences of voltage values $\{V_{a,n}\} = \{V_a(T_n)\}$, $a \in A$, are computed sequentially via

$$V_{a,n} - I_a/G_a = \left( V_{a,n-1} - I_a/G_a \right) e^{-\frac{T_n - T_{n-1}}{\tau}} + J_{a,n}, \quad a \in A.$$

Finally, consider a simulation period $T$ for which there are $N$ synaptic event times $T_n$, $1 \leq n \leq N$, and with the convention $T_0 = 0$ and $T_{N+1} = T$. Given $A_m$, a multiset of elements of $A$ such that $|A_m| = m$, the corresponding $m$-th order empirical stationary moment is evaluated as

$$
\begin{aligned}
\mu_{A_m,T} &= \frac{1}{T} \int_0^T \prod_{a \in A_m} \left( V_a(t) - I_a/G_a \right) \mathrm{d}t, \\
&= \frac{1}{T} \sum_{n=0}^{N} \int_{T_n}^{T_{n+1}} \prod_{a \in A_m} \left( V_a(t) - I_a/G_a \right) \mathrm{d}t, \\
&= \frac{1}{T} \sum_{n=0}^{N} \left( \int_{T_n}^{T_{n+1}} e^{-\frac{m(t-T_n)}{\tau}} \mathrm{d}t \right) \prod_{a \in A_m} \left( V_{a,n} - I_a/G_a \right), \\
&= \frac{\tau}{mT} \sum_{n=0}^{N} \left( 1 - e^{-\frac{m(T_{n+1}-T_n)}{\tau}} \right) \prod_{a \in A_m} \left( V_{a,n} - I_a/G_a \right).
\end{aligned}
$$

## Results

Cortical activity typically exhibits a high degree of variability in response to identical stimuli [62,63], with individual neuronal spiking exhibiting Poissonian characteristics [64,65]. Such variability is striking because neurons are thought to receive large numbers ($\simeq 10^4$) of synaptic contacts [66]. Given such large numbers, in the absence of synchrony, neuronal variability should average out, leading to quasi-deterministic neuronal voltage dynamics [67]. However, contrary to this prediction, electrophysiological *in-vivo* measurements reveal that

neuronal membrane voltage exhibits fluctuations with typical variance values of $\simeq 4 - 9\text{mV}^2$ [3,4]. Note that these variance estimates are obtained from voltage traces where action potentials have been excluded, so that the measured variability can be primarily attributed to synaptic inputs. In our previous work [6], we argue that achieving physiological cortical variability requires input synchrony within the AONCB modeling framework. We summarize these results in Fig 5, where we examine three distinct conditions of input synchrony: asynchronous input, separately synchronous excitatory and inhibitory inputs, and jointly synchronous excitatory and inhibitory inputs.

In this work, we extend our prior results derived in [6] to the more general setting of neuronal feedforward populations driven by heterogeneously synchronous inputs, but still under the assumption of instantaneous synapses. In this more general setting, we then ask whether realistic level of synchrony impact other important aspect of the subthreshold variability, specifically voltage covariability across neurons and voltage skewness in individual neurons.

## Voltage covariance

In order to specify the voltage covariance of synchronously driven AONCB neurons, one must first evaluate their stationary voltage mean. Applying the PASTA principle to a single AONCB

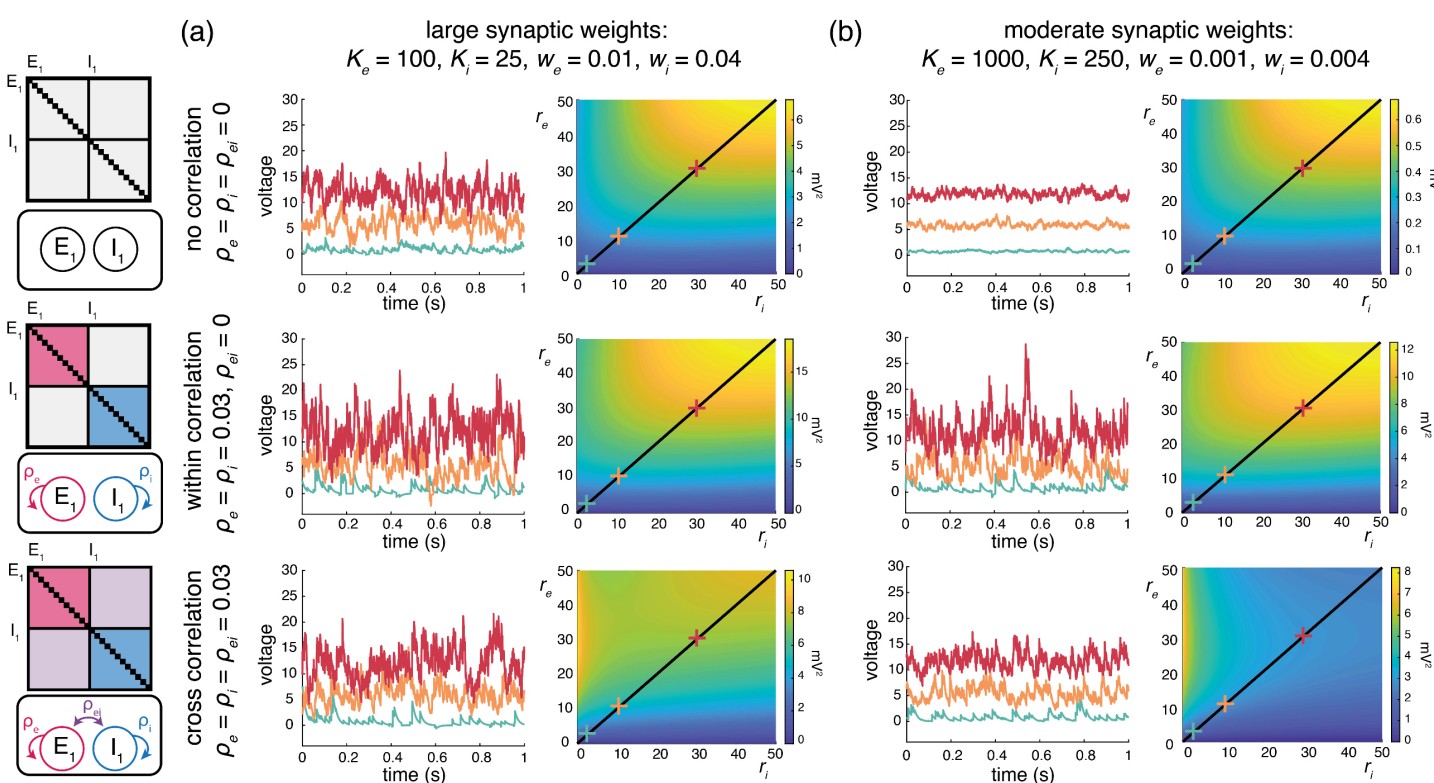

**Fig 5. Voltage variance depends on input strength and correlations.** (a) Example Monte Carlo simulations of AONCB neurons with large synaptic weights at various input firing rates (blue, 1Hz; orange, 10 Hz; red, 30 Hz) for different correlation conditions (top, asynchronous; middle, within-pool correlations; bottom, across-pool correlations). Voltage variance as a function of rate can be seen on the right with same correlation conditions. (b) Same as (a) but for moderately sized synaptic weights.

neuron labeled by 1 in S1 Text, S2 Appendix, we find this mean to be

$$m_1 = \frac{b_1 \mathbb{E}\left[ \frac{W_{e,1} V_e + W_{i,1} V_i}{W_{e,1} + W_{i,1}} \left(1 - Y_1\right) \right] + I_1/(\tau_1 G_1)}{1/\tau_1 + b_1 \mathbb{E}\left[1 - Y_1\right]},$$

where the auxiliary random variable $Y_1$ is defined as $Y_1 = e^{-(W_{e,1} + W_{i,1})}$. Note that in the expression above, $b_1$ is the rate of the compound Poisson process modeling the synaptic inputs to neuron 1 and that the expectation is with respect to the jump distribution of $(W_{e,1}, W_{i,1})$. The expression of the stationary mean voltage can be conveniently recast as

$$m_1 = \frac{c_{e,1} V_{e,1} + c_{i,1} V_{i,1} + I_1/(b_1 \tau_1 G_1)}{1/\tau_1 + c_{e,1} + c_{i,1}}, \tag{14}$$

where the rate coefficients $c_{\alpha,1}, \alpha \in \{e, i\}$ are given by

$$c_{\alpha,1} = b_1 \mathbb{E}\left[ \frac{W_{\alpha,1}}{W_{\alpha,1} + W_{\alpha,1}} \left(1 - e^{-\sum_{\alpha \in \{e,i\}} W_{\alpha,1}}\right) \right]. \tag{15}$$

Eq (14) has superficially the same form as for deterministic rate dynamics with constant conductances, in the sense that the mean voltage is a weighted sum of the reversal potentials $V_e$, $V_i$ and $V_L = 0$. One can check that for such deterministic rate dynamics, the synaptic efficacies involved in the stationary mean simply read $c_{\alpha,1} = \sum_{k=1}^{K_\alpha} r_{\alpha,k} w_{\alpha,1,k}$. By contrast, modeling synaptic conductances as asynchronous shot noise leads to synaptic efficacies under exponential form: $c_{\alpha,1} = \sum_{k=1}^{K_\alpha} r_{\alpha,k}(1 - e^{-w_{\alpha,1,k}})$. In turn, accounting for input synchrony leads to synaptic efficacies expressed as expectation of these exponential forms, as in Eq (15), consistent with the fact that our approach models synchrony via the stochastic nature of the conductance jumps $(W_e, W_i)$.

Applying the PASTA principle to a pair of AONCB neurons labeled by 1 and 2, we find in S1 Text, S5 Appendix, that the stationary covariance of their voltages is given as

$$M_{V_1, V_2} = \frac{b_{12} \mathbb{E}\left[ \left( \frac{W_{e,1} V_e + W_{i,1} V_i}{W_{e,1} + W_{i,1}} - m_1 \right) \left(1 - Y_1\right) \left( \frac{W_{e,2} V_e + W_{i,2} V_i}{W_{e,2} + W_{i,2}} - m_2 \right) \left(1 - Y_2\right) \right]}{1/\tau_1 + 1/\tau_2 + b_{12} \mathbb{E}\left[1 - Y_1 Y_2\right]}, \tag{16}$$

where the auxiliary random variables $Y_a$, $a = 1, 2$, satisfy $Y_a = e^{-(W_{e,a} + W_{i,a})}$. The above expression calls for a few observations: First, note that $b_{12}$ denotes the rate of the compound Poisson process modeling the synaptic inputs to the pair of neurons. In particular, we have $b_1, b_2 \leq b_{12} \leq b_1 + b_2$. Second, note that the expectation is evaluated with respect the jump probability of $(W_{e,1}, W_{i,1}, W_{e,2}, W_{i,2})$. Third, note that Eq (16) is consistent with the expression for the voltage variance of an AONCB neuron derived in [6], which is recovered by setting $V_1 = V_2$. Similarly as for the stationary mean, the expression of the stationary voltage covariance can be conveniently recast as

$$M_{V_1, V_2} = \frac{\sum_{\alpha, \beta \in \{e,i\}} c_{\alpha\beta,12}(V_\alpha - m_1)(V_\beta - m_2)}{1/\tau_1 + 1/\tau_2 + \sum_{\alpha \in \{e,i\}} (c_{\alpha,1} + c_{\alpha,2}) - \sum_{\alpha, \beta \in \{e,i\}} c_{\alpha\beta,12}}. \tag{17}$$

where the rate coefficients $c_{\alpha\beta,12}$, $\alpha, \beta \in \{e, i\}$ are given by

$$c_{\alpha\beta,12} = b_{12}\mathbb{E}\left[W_{\alpha,1}W_{\beta,2}\frac{\left(1 - e^{-(W_{e,1}+W_{i,1})}\right)\left(1 - e^{-(W_{e,2}+W_{i,2})}\right)}{(W_{e,1}+W_{i,1})(W_{e,2}+W_{i,2})}\right]. \tag{18}$$

Before discussing the implications of input synchrony across neurons, we discuss the implication of the results presented above for the voltage variance.

### Impact of synchrony on individual voltage variability

In the framework considered here, the voltage variance $M_{V_1,V_1}$ for neuron 1 is obtained by considering Eqs (16) and (17) under the assumption that neurons 1 and 2 are exchangeable and share the same inputs with identical weights. In particular, this corresponds to $\tau_1 = \tau_2$, $b_{12} = b_1 = b_2$, and $W_{\alpha,1} = W_{\alpha,2}$. We devote the next three subsections to discuss the variance formula for $M_{V_1,V_1}$ under various assumption of synchrony.

**Fully asynchronous inputs.** Let us first consider the purely asynchronous case for which each synaptic input behaves independently. In the context of our jump-process-based model for the synaptic drive, this means that the rate of the governing Poisson process is the sum of the individual synapses' input rates, i.e.,

$$b_1 = \sum_{\alpha \in \{e,i\}} \sum_{k=1}^{K_\alpha} \mathbb{1}_{\{w_{\alpha,1,k}>0\}} r_{\alpha,k}. \tag{19}$$

Independence of the synaptic inputs also means that every synaptic event comprises a single active synapse and that the probability that a particular input is active is determined in keeping to its input rate. For instance, the probability that the $k$-th synapse of type $\alpha \in \{e, i\}$ is the active one is $\mathbb{P}\left[X_{\alpha,k} = 1, X_{\alpha,l\neq k} = 0\right] = r_{\alpha,k}/b_1$. Next, by independence between excitation and inhibition, we have $c_{\alpha\beta,11} = 0$ whenever $\alpha \neq \beta$. We also have $W_{\alpha,1}/(W_{e,1}+W_{i,1}) = \mathbb{1}_{\{W_{\alpha,1}>0\}}$ since for all synaptic events, either $W_{e,1} = 0$ or $W_{i,1} = 0$. Then evaluating the expectation in Eq (18) with respect to the monosynaptic activation probability $\mathbb{P}\left[X_{\alpha,k} = 1, X_{\alpha,l\neq k} = 0\right] = r_{\alpha,k}/b$ leads to $c_{\alpha\alpha,11} = \sum_{k=1}^{K_\alpha} r_{\alpha,k}\left(1 - e^{-w_{\alpha,1,k}}\right)^2$. As a result, Eq (17) reads

$$M_{V_1,V_1}\Big|_{\rho_1=0} = \frac{\sum_{k=1}^{K_e} r_{e,k}(1 - e^{-w_{e,1,k}})^2(V_e - m_1)^2 + \sum_{l=1}^{K_i} r_{i,l}(1 - e^{-w_{i,1,l}})^2(V_i - m_1)^2}{2/\tau_1 + \sum_{k=1}^{K_e} r_{e,k}(1 - e^{-2w_{e,1,k}}) + \sum_{l=1}^{K_i} r_{i,k}(1 - e^{-2w_{i,1,l}})}.$$

The above result coincides with the expression obtained in the effective-time-constant approximation [53], except for the exponential form of the synaptic efficacies. As stated earlier, this exponential form is due to the shot-noise nature of the drive, and one can recover the classically derived expression by making the additional assumption of small synaptic weight, i.e., $w_{e,k}, w_{i,l} \ll 1$, for which we have:

$$M_{V_1,V_1}\Big|_{\rho_1=0} \simeq \frac{\sum_{k=1}^{K_e} r_{e,k}w_{e,1,k}^2(V_e - m_1)^2 + \sum_{l=1}^{K_i} r_{i,l}w_{i,1,l}^2(V_i - m_1)^2}{2\left(1/\tau_1 + \sum_{k=1}^{K_e} r_{e,k}w_{e,1,k} + \sum_{l=1}^{K_i} r_{i,k}w_{i,1,l}\right)}.$$

**Pool-specific synchrony.** Suppose that neuron 1 receives inputs that only exhibit synchrony when considered separately as a pool of excitatory or inhibitory inputs. We still have $b_1 = b_{e,1} + b_{i,1}$ but in the presence of within-pool synchrony, the rates $b_{\alpha,1}$ are only subadditive

functions of the synaptic rates because inputs can coactivate within the pool:

$$\max_{w_{\alpha,k,1}>0} r_{\alpha,k} \leq b_{\alpha,1} < \sum_{k=1}^{K_\alpha} \mathbb{1}_{\{w_{\alpha,1,k}>0\}} r_{\alpha,k}. \tag{20}$$

Further specifying a functional form for $b_{\alpha,1}$ requires adopting a parametric model for synchrony, e.g., derived from the beta-binomial statistical model as in [6]. However, even without choosing a parametric model, one can always assume that $b_1$ is known as it is entirely specified by Eqs (8) and (9). The presence of pool-specific synchrony means that the number of coactivating inputs may vary across synaptic events. Consequently, the rate coefficients given by Eq (18) shall be evaluated as true expectations

$$c_{\alpha\alpha,11} = b_1 \mathbb{E}\left[\left(1 - e^{-W_{\alpha,1}}\right)^2\right] \quad \text{with} \quad W_{\alpha,1} = \sum_{k=1}^{K_\alpha} X_{\alpha,k} w_{\alpha,k,1},$$

where the randomness of the jump $W_{\alpha,1}$ is the hallmark of pool-specific synchrony. Although $W_{\alpha,1}$ can be much larger than any individual synaptic weights, for small enough synaptic weights and weak enough synchrony, the small-weight approximation still holds with large probability: $\mathbb{P}\left[W_{\alpha,1} \ll 1\right] \simeq 1$. Under the small-weight approximation, one can express the rate coefficients $c_{\alpha\alpha,11}$ directly in terms of the synaptic activation variables

$$c_{\alpha\alpha,11} \simeq \sum_{1 \leq k,l \leq K_\alpha} w_{\alpha,a,k} w_{\alpha,a,l} b_1 \mathbb{E}\left[X_{\alpha,k} X_{\alpha,l}\right].$$

In turn, remembering that by definition, we have $b_1 \mathbb{E}\left[X_{\alpha,k} X_{\beta,l}\right] = \rho_{\alpha\beta,kl}\sqrt{r_{\alpha,k} r_{\beta,l}}$, the variance with pool-specific synchrony reads

$$M_{V_1,V_1}\Big|_{\rho_{ei,1}=0} \simeq \frac{\sum_{\alpha\in\{e,i\}} (V_\alpha - m_1)^2 \sum_{1\leq k,l\leq K_\alpha} \rho_{\alpha\alpha,kl}\sqrt{r_{\alpha,k}r_{\alpha,l}} w_{\alpha,1,k} w_{\alpha,1,l}}{2\left(1/\tau_1 + \sum_{\alpha\in\{e,i\}} \sum_{k=1}^{K_\alpha} r_{\alpha,k} w_{\alpha,1,k}\right)}. \tag{21}$$

As the diagonal correlation coefficients are unit value, i.e., $\rho_{\alpha\alpha,kk} = 1$, one can see that pool-specific synchrony can only increase voltage variability via the remaining off-diagonal terms $\rho_{\alpha\alpha,kl}$, $k \neq l$. Moreover, assuming uniform spiking correlations $\rho_{\alpha\alpha,kl} = \rho_{\alpha,1}$, $k \neq l$, uniform synaptic weights $w_{\alpha,1,k} = w_{\alpha,1}$, and uniform input rates $r_{\alpha,k} = r_\alpha$, one obtains

$$M_{V_1,V_1}\Big|_{\rho_{ei,1}=0} \simeq \frac{\sum_{\alpha\in\{e,i\}} (V_\alpha - m_1)^2 K_{\alpha,1} r_\alpha \left(1 + (K_{\alpha,1} - 1)\rho_{\alpha,1}\right) w_{\alpha,1}^2}{2\left(1/\tau_1 + \sum_{\alpha\in\{e,i\}} K_{\alpha,1} r_\alpha w_{\alpha,1}\right)},$$

where $K_{\alpha,1}$ is the number of synaptic input received by neuron 1 and $\rho_{\alpha,1}$ is the spiking correlation between synaptic inputs of type $\alpha$ impinging on neuron 1. This shows that synchrony significantly impacts voltage variability as soon as $K_{\alpha,1} \geq 1/\rho_{\alpha,1} \geq 100$, which generally holds in cortex [66].

**Synchrony between excitation and inhibition.** Let us finally consider the impact of having synchrony between excitation and inhibition on the variance $M_{V_1,V_1}$, for which excitatory and inhibitory synapses may coactivate. As a result of such synchrony, the rate $b_1$ is no longer additive: $b_1 < b_{e,1} + b_{i,1}$. Similarly, it no longer holds that $W_{\alpha,1}/(W_{e,1} + W_{i,1}) = \mathbb{1}_{\{W_{\alpha,1}>0\}}$ and one cannot simplify the fractional form of Eqs (15) and (18). This fractional form can

be understood as a mitigation of the antagonistic forces exerted by excitation and inhibition during a mixed synaptic event and closely mirrors the form of the Marcus rule update in Eq (7). Thus, synchrony between excitation and inhibition alters the expression of all coefficients $c_{\alpha\beta,11}$, $\alpha, \beta \in \{e, i\}$, and in particular, it causes the cross-coefficient $c_{ei,11}$ to be positive. Accordingly, in the small-noise approximation, one obtains

$$M_{V_1,V_1} \simeq \frac{\sum_{\alpha,\beta\in\{e,i\}} (V_\alpha - m_1)(V_\beta - m_1) \sum_{k=1}^{K_\alpha} \sum_{l=1}^{K_\beta} \rho_{\alpha\beta,kl} \sqrt{r_{\alpha,k} r_{\beta,l}} w_{\alpha,1,k} w_{\beta,1,l}}{2\left(1/\tau_1 + \sum_{\alpha\in\{e,i\}} \sum_{k=1}^{K_e} r_{\alpha,k} w_{\alpha,1,k}\right)}.$$

The above expression only differs from Eq (21) by the presence of cross terms involving $\rho_{ei,kl} (V_e - m_1)(V_i - m_1)$. The latter quantity is necessarily negative as the voltage $V$ is bounded above and below by $V_e$ and $V_i$, respectively. Thus, the impact of synchrony between excitation and inhibition is to reduce variability in the membrane voltage, as intuition suggests [68]. Under assumptions of uniformity about spiking correlation coefficients, synaptic weights, and input rates, we further have

$$M_{V_1,V_1} \simeq M_{V_1,V_1}\big|_{\rho_{ei,1}=0} - \frac{(V_e - m_1)(m_1 - V_i) \rho_{ei,1} K_{e,1} K_{i,1} \sqrt{r_{e,1} r_{i,1}} w_{e,1} w_{i,1}}{1/\tau_1 + \sum_{\alpha\in\{e,i\}} K_{\alpha,1} r_\alpha w_{\alpha,1}}.$$

Observe that the uniform spiking correlation between excitatory and inhibitory inputs to neuron 1, denoted above by $\rho_{ei,1}$, necessarily satisfies $\rho_{ei,1} \le \sqrt{\rho_{e,1}\rho_{i,1}}$, so that as expected, the variance remains a nonnegative number.

## Impact of synchrony on voltage covariability

Simultaneous whole-cell measurements in pairs of neighboring cells have revealed highly synchronized membrane voltage recordings, with crosscorrelation coefficients as large as $\rho \simeq 0.8$ [17]. Because these correlation measurements are essentially independent of the occurrence of spikes, their high levels of synchrony are almost entirely attributable to subthreshold activity. Simultaneous pair recordings in the voltage-clamp configuration further showed that similarly high levels of synchrony hold among excitatory and inhibitory inputs, as well as across excitatory and inhibitory inputs [19]. Although the nature of synchrony may be altered by stimulus drive, which can shift synchrony to higher-frequency voltage fluctuations, the overall degree of synchrony across nearby cells has consistently been measured to be high in anesthetized animals [20]. These findings were confirmed in awake behaving animals, where simultaneous pair recordings yield crosscorrelation coefficients $\rho \simeq 0.8$ during spontaneous, restful activity, and comparatively weaker albeit still large values $\rho \simeq 0.6$ during sensory drive or motor activity [18,21].

In this work, we consider whether the level of spiking synchrony necessary to explain voltage variability in single neurons can also explain the high degree of covariability observed between neighboring cells. To address this question within our modeling framework, we now discuss Eq (16) in the presence of synchrony between inputs to neuron 1 and neuron 2, which generally implies nonzero voltage covariance: $M_{V_1,V_2} \ne 0$.

**Synchrony between inputs to distinct neurons.** By contrast with the variance case, evaluating $M_{V_1,V_2}$ via Eq (16) requires knowledge of the input rate to the neuronal pair $b_{12}$ as opposed to the individual neuronal input rates $b_1$ and $b_2$. In S1 Text, S6 Appendix, we show

that the pair-specific input rate satisfies

$$b_{12} = \frac{b_1 + b_2}{1 + q_{12}} \quad \text{with} \quad q_{12} = \mathbb{P}\left[W_1 > 0, W_2 > 0 \mid W_1 + W_2 > 0\right], \tag{22}$$

where we have denoted by $W_a = W_{e,a} + W_{i,a}$ the aggregate jump experienced by neuron $a$. The term $q_{12}$ is the probability that neurons 1 and 2 receive synchronous inputs given that at least one neuron of the pair receives an input. Specifying a functional form for $q_{12}$ requires choosing a parametric model for synchrony, i.e., for the jump distribution $(W_{e,a}, W_{i,a})_{a\in\{1,2\}}$. Given such a model, one can use Eq (22) to relate the value of $b_{12}$ to $b_1$ and $b_2$, which in turn, can be deduced from the individual synaptic rates (see S1 Text, S6 Appendix). When inputs to neurons 1 and 2 are independent, we have $q_{12} = 0$ so that $b_{12} = b_1 + b_2$, whereas when neurons are optimally synchronous, we have $q_{12} = \min(b_1, b_2)/\max(b_1, b_2)$, so that $b_{12} = \max(b_1, b_2)$. For intermediary level of synchrony between inputs to neurons 1 and 2, one generally has $\max(b_1, b_2) < b_{12} < b_1 + b_2$. At the same time, it no longer necessarily holds that $(1 - Y_1)(1 - Y_2) = 0$, or equivalently that $W_{\alpha,1}W_{\beta,2} = 0$ for all $\alpha, \beta \in \{e, i\}$. As a result, all four cross-coefficients $c_{\alpha\beta,12}$ can be positive in Eq (18), depending on whether there is synchrony in between various neuron-specific pools of excitatory and inhibitory inputs. The fact that $c_{\alpha\beta,12} \neq 0$, as well as the fact that $b_{12} < b_1 + b_2$, are the hallmark of the presence of synchrony between inputs to neuron 1 and 2.

As in the case of the voltage variance, considering the biophysically relevant small-weight approximation yields directly interpretable formulas. These formulas are interpretable in as much as they only involve the input spiking rates $r_{\alpha,k}$ and the spiking correlation coefficients $\rho_{\alpha\beta,kl}$. Specifically, in the small-weight approximation, one obtains the generic covariance formula

$$M_{V_1,V_2} \simeq \frac{\sum_{\alpha,\beta\in\{e,i\}}(V_\alpha - m_1)(V_\beta - m_2)\sum_{k=1}^{K_\alpha}\sum_{l=1}^{K_\beta}\rho_{\alpha\beta,kl}\sqrt{r_{\alpha,k}r_{\beta,l}}w_{\alpha,1,k}w_{\beta,2,l}}{1/\tau_1 + 1/\tau_2 + \sum_{\alpha\in\{e,i\}}\sum_{k=1}^{K_\alpha}r_{\alpha,k}(w_{\alpha,1,k} + w_{\alpha,2,k})}.$$

where $\rho_{\alpha\beta,kl}$ denotes the spiking correlation coefficients between inputs of type $\alpha$ and inputs of type $\beta$ to distinct neurons. Under assumptions of uniformity about spiking correlation coefficients, synaptic weights, and input rates, we further have

$$M_{V_1,V_2} \simeq \frac{\sum_{\alpha,\beta\in\{e,i\}}(V_\alpha - m_1)(V_\beta - m_2)\rho_{\alpha\beta,12}K_\alpha K_\beta\sqrt{r_\alpha r_\beta}w_{\alpha,1}w_{\beta,2}}{1/\tau_1 + 1/\tau_2 + \sum_{\alpha\in\{e,i\}}K_\alpha r_\alpha(w_{\alpha,1} + w_{\alpha,2})}, \tag{23}$$

which shows that although spiking correlation coefficients must be nonnegative within our framework, one can have $M_{V_1,V_2} < 0$ if, e.g., one assumes $\rho_{ee,12} = \rho_{ii,12} = 0$ and $\rho_{ei,12} = \rho_{ie,12} > 0$. As for the voltage variance, the small-weight approximation of the covariance $M_{V_1,V_2}$ takes a simple interpretable quotient form. Specifically, the numerator expression has the same quadratic form as the one obtained for the voltage covariance of current-based models, which is recovered by neglecting the rate-dependent component of the denominator. Moreover, just as for the voltage variance, the covariance for conductance-based models follows from normalization by an effective time constant, which is obtained as the harmonic mean of the effective neuron-specific time constants. Finally, observe that if the covariance depends nonlinearly on the input rates $r_{\alpha,k}$ and the synaptic weights $w_{\alpha,1,k}$ and $w_{\alpha,2,k}$, it is a linear function of the spiking correlation coefficients $\rho_{\alpha\beta,kl}$.

**Shared asynchronous inputs.** We first consider under which conditions realistic levels of voltage variability and covariability may emerge in the absence of input synchrony. Our previous work supports that achieving realistic voltage variability in the absence of input synchrony is possible when driven by a relatively small number ($K_e = 4K_i \simeq 100$) of strong synaptic inputs ($w_e = w_i/4 \simeq 0.01$). Given this asynchronous setting, neuronal voltages may still covary across pairs of neurons if they share a fraction of their synaptic inputs. The case of shared inputs corresponds to considering distinct inputs of the same type $\alpha$ to neuron 1 and 2, say input $k$ to 1 and input $l$ to 2, but with identical rates and unit pairwise spiking correlation coefficient: $r_{\alpha,k} = r_{\alpha,l}$ and $\rho_{\alpha\alpha,kl} = 1$. For simplicity, we consider the symmetric case for which two identical neurons 1 and 2 receive the same number $K_e$ and $K_i$ of synaptic inputs via uniform synaptic weights $w_e$ and $w_i$ and uniform firing rate $r_e$ and $r_i$, while sharing a fraction $f_e$ of the excitatory inputs and a fraction $f_i$ of inhibitory inputs. In this symmetric case, the voltage correlation between neurons is given by

$$\rho_{V_1,V_2} = \frac{M_{V_1,V_2}}{\sqrt{M_{V_1,V_1}M_{V_2,V_2}}} = f_e q + f_i(1-q),$$

where $q$ measures the relative share of variability due to excitation as opposed to inhibition

$$q = \frac{K_e r_e w_e^2 (V_e - m)^2}{\sum_{\alpha \in \{e,i\}} K_\alpha r_\alpha w_\alpha^2 (V_\alpha - m)^2}.$$

When excitation alone is considered, we simply have that $\rho_{V_1,V_2} = f_e$ as intuition suggests. Given that $V_e \simeq 60\text{mV} > V_i \simeq -10\text{mV}$, $q$ is defined as a decreasing function of the mean voltage $m$ in the presence of both excitation and inhibition. To estimate $q$, we consider biophysically realistic parameter values for which $K_e w_e, K_i w_i \simeq 1$ and $w_i/w_e \simeq 4$ and we assume that excitatory and inhibitory rates are similar: $r_e \simeq r_i$. In this setting, one can check that excitation is responsible for 90% of the variability at resting potential $m = 0$, and that this share declines to about 45% for $m = 15\text{mV}$ depolarization. Thus, to achieve $\rho_{V_1,V_2} \simeq 0.6 - 0.8$ during spontaneous activity and $\rho_{V_1,V_2} \simeq 0.4 - 0.6$ in the the driven regime, one must assume that both neurons share between 65% – 85% of their excitatory inputs and between 20% – 40% of their inhibitory inputs. Both proportions of shared inputs, especially for the excitatory inputs, are larger than those typically observed experimentally, with estimates ranging from as low as 5% [9] to at most 30% shared inputs [69,70]. This suggests that neurons driven by a relatively low number of large asynchronous synapses can exhibit realistic levels of voltage variance but fail to produce realistic levels of voltage covariability.

We illustrate these results for asynchronous input drives in Fig 6. In Fig 6a–6b, we emphasize that the voltage covariability mostly depends (quasi-linearly) on the fraction of shared excitatory inputs $f_e$ as opposed to the number of excitatory inputs $K_e$. This is by contrast with Fig 6c–6d which shows that voltage covariability only marginally depends on the fraction of shared inhibitory inputs $f_i$, and even less on the number of inhibitory inputs $K_i$. However, in both cases, at fixed $f_e$ and $f_i$, increasing the synaptic connectivity numbers $K_e$ and $K_i$ impacts the individual neurons statistics, and in particular, leads to voltage variance increases as observed *in-vivo* [3,4,19]. The primary dependence of voltage covariability on the shared fraction of excitatory inputs follows from the fact that for biophysically relevant parameters, excitatory inputs are responsible for most of the voltage variability, yielding near unit values for $q$. This is because at physiological level of depolarization, the membrane voltage sits much further from the excitatory reversal potential $V_e$ than from the inhibitory reversal potential $V_i$, leading to a comparatively larger excitatory driving force. Finally, in Fig 6e–6f, we confirm

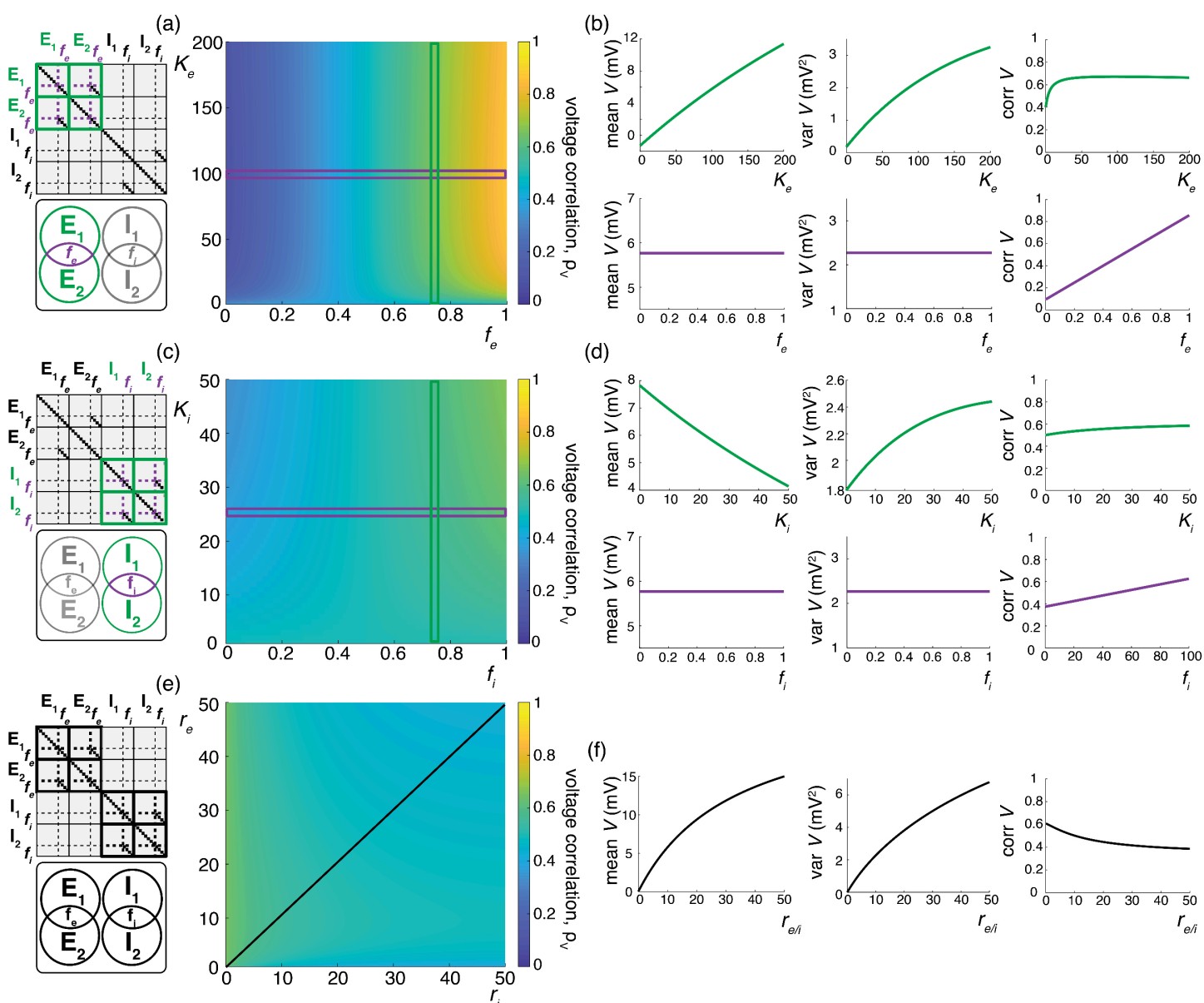

**Fig 6. Synaptic effects on voltage correlations through asynchronous shared input with large synaptic weights.** (a-b) Changes in neural statistics in response to a varying excitatory drive with $r_e$ = 10 Hz, while inhibition is held constant at $K_i$ = 25, $r_i$ = 10 Hz, and $f_i$ = 75%. (a) Voltage correlation as a function of excitatory inputs $K_e$ and the percent of $K_e$ shared between the two neurons $f_e$. (b) Changes in voltage statistics (mean, variance, and voltage correlation) while holding either a constant percent shared $f_e$ at 75% (top, green) or a constant $K_e$ of 100 (bottom, purple). (c-d) Changes in neural statistics in response to a varying inhibitory drive with $r_i$ = 10 Hz, while excitation is held constant at $K_e$ = 100, $r_e$ = 10 Hz, and $f_e$ = 75%. (c) Voltage correlation as a function of inhibitory inputs $K_i$ and the percent of $K_i$ shared between the two neurons $f_i$. (d) Changes in voltage statistics (mean, variance, and voltage correlation) while holding either a constant percent shared $f_i$ at 75% (top, green) or a constant $K_i$ of 25 (bottom, purple). (e-f) Changes in neural statistics in response to increasing drive with $K_e$ = 200, $f_e$ = 0.85%, $K_i$ = 50 and $f_i$ = 0.4%. (e) Voltage correlation as a function of input rate $r_e$ and $r_i$ in Hz. (f) Changes in voltage statistics (mean, variance, and voltage correlation) while jointly increasing both the excitatory and inhibitory drive $r_{e/i}$. In all cases the synaptic weights are large with $4W_e = W_i = 0.04$.

that albeit unrealistic, assuming shared fractions $f_e$ = 85% and $f_i$ = 40% leads to physiologically plausible individual and joint neural responses, with voltage correlations $\rho_{V_1,V_2} = 0.6$ at resting state and $\rho_{V_1,V_2} = 0.4$ in the driven regime.

**Synchronous inputs across neurons.** We next consider under which conditions realistic levels of voltage variability and covariability may emerge in the presence of physiological input synchrony. This corresponds to spiking correlation coefficients within the range $\rho = 0.01 – 0.04$. Our previous work supports that achieving realistic voltage variability in the presence of input synchrony is possible when driven by a relatively large number ($K_e = 4K_i \simeq 10^4$) of moderate synaptic inputs ($w_e = w_i/4 \simeq 0.001$). Given this synchronous setting, we consider the case of two identical neurons driven by synchronous synaptic inputs with symmetric, uniform spiking correlation structure. Moreover, we consider that when considered separately, inputs to each neuron exhibit a similar degree of spiking correlations, whereas inputs to separate neurons are allowed to exhibit a distinct degree of spiking correlations. Specifically, we consider two such correlation structures: In case ($i$), for ease of comparison with the case of shared independent inputs, we consider that excitatory and inhibitory inputs are independent so that $\rho_{\alpha\beta,11} = \rho_{\alpha\beta,22} = \rho_{\alpha\beta,12} = 0$ and the correlations are parametrized by $\rho_{\alpha\alpha,11} = \rho_{\alpha\alpha,22} = \rho > \rho' = \rho_{\alpha\alpha,12} = \rho_{\alpha\alpha,12}$ for all $\alpha \in \{e, i\}$. In case ($ii$), we consider the more physiologically relevant case of uniformly correlated excitatory and inhibitory inputs for which $\rho_{\alpha\beta,11} = \rho_{\alpha\beta,22} = \rho$ for all $\alpha, \beta \in \{e, i\}$ and such that for all $\alpha, \beta \in \{e, i\}$ with $\alpha \neq \beta$, $\rho_{\alpha\beta,12} = \rho_{\alpha\alpha,12} = \rho' < \rho$. For both cases ($i$) and ($ii$), the individual voltage variance can be written as the weighted average of the asynchronous variance $M_{V_1,V_1}|_{\rho=0}$ and the fully synchronous variance $M_{V_1,V_1}|_{\rho=1}$: $M_{V_1,V_1} = (1-\rho)M_{V_1,V_1}|_{\rho=0} + \rho M_{V_1,V_1}|_{\rho=1}$. Similarly, the co-variance depends linearly on the spiking correlation $\rho'$ so that $M_{V_1,V_2} = \rho' M_{V_1,V_2}|_{\rho'=1}$ where the fully synchronous covariance is such that $M_{V_1,V_2}|_{\rho'=1} = M_{V_1,V_1}|_{\rho=1}$. Therefore, in the symmetric case, the expression of the voltage correlations between neuron 1 and 2 simplifies to

$$\rho_{V_1,V_2} = \frac{M_{V_1,V_2}}{\sqrt{M_{V_1,V_1} M_{V_2,V_2}}} = \frac{\rho'}{(1-\rho)/\kappa + \rho} < \frac{\rho'}{\rho} \leq 1, \tag{24}$$

where the dimensionless parameter $\kappa$ measures the ratio of the fully synchronous variance to the asynchronous variance:

$$\kappa = \frac{M_{V_1,V_1}|_{\rho=1}}{M_{V_1,V_1}|_{\rho=0}} = \begin{cases} K_e q + K_i(1-q) & \text{in case } (i), \\ \left(\sqrt{K_e q} + \sqrt{K_i(1-q)}\right)^2 & \text{in case } (ii). \end{cases} \tag{25}$$

When excitation alone is considered, i.e., for $q = 1$, the above expression reduces to $\kappa = K_e$ and voltage correlations exhibit two regimes: for small connectivity number $K_e < 1/\rho_e = 25 – 100$, $\rho_{V_1,V_2}$ is linear function of $K_e$ with $\rho_{V_1,V_2} = \rho' K_e$; for large connectivity number $K_e > 1/\rho_e = 25 – 100$, $\rho_{V_1,V_2}$ saturates to $\rho_{V_1,V_2} = \rho'/\rho$. Thus realistic level of correlations are achieved for relatively large connectivity numbers if one assumes weaker spiking correlation across neuronal inputs than within neuron-specific inputs so that $\rho'/\rho \simeq 0.6 – 0.8$. Assuming that excitatory and inhibitory rates are similar, i.e., $r_e \simeq r_i$, the parameter $\kappa$, and thus the coefficient $\rho_{V_1,V_2}$, become nonmonotonic functions of the mean depolarization $m$, which are however both decreasing over the physiologically range for which $-10\text{mV} \leq m \leq 20\text{mV}$. For these values, one can check that including inhibition reduces the value of $\kappa$ to 60% of its value at resting potential $m = 0$ (i.e. $\kappa \simeq 0.6K_e$) and that this value further declines to about 8% for $m = 15\text{mV}$ depolarization (i.e. $\kappa \simeq 0.08K_e$). Such a reduction is due to the fact that including synchronous inhibition cancels out part of the variability due to synchronous excitatory inputs, a cancellation effect that is magnified for larger depolarizations, as the inhibitory driving force increases. In turn, given these estimates for $\kappa$ with $K_e \simeq 10^4$, choosing the biophysically realistic values $\rho' = 0.013 – 0.025 < \rho = 0.02 – 0.03$ yields $\rho_{V_1,V_2} \simeq 0.6 – 0.8$ during

spontaneous activity and $\rho_{V_1,V_2} \simeq 0.4 – 0.6$ in the the driven regime, as observed physiologically. This suggests that including the weak level of synchrony observed in the input drive is enough to jointly account for the voltage variability and covariability of the subthreshold dynamics of neuronal pairs.

We illustrate the impact of synchrony on voltage covariability discussed above in Fig 7. To compare this impact with that of shared independent inputs, we first examine case (*i*), for which synchronous excitation and synchronous inhibition act independently. Similarly to the case of shared inputs, Fig 7a–7b, emphasizes that the voltage covariability primarily depends (linearly) on the cross excitatory correlations, $\rho'_e$, as opposed to the number of excitatory inputs $K_e$. This is by contrast with Fig 7c–7d which shows that voltage covariability only marginally depends on the cross inhibiotry correlations, $\rho'_i$, and even less on the number of inhibitory inputs $K_i$.

Finally, assuming $r_e \simeq r_i$, we confirm in Fig 7f that including physiological levels of input synchrony yields realistic voltage correlations at resting state ($\rho_{V_1,V_2} = 0.6$ for $r_e, r_i \simeq 1Hz$) as well as in the driven regime ($\rho_{V_1,V_2} = 0.4$ for $r_e, r_i \simeq 50Hz$). These results indicate that assuming broad level of input synchrony, compatible with the weak level of spiking correlations observed *in-vivo*, is enough the explain the high degree of voltage correlations observed in simultaneous pair recordings.

## Voltage skewness

In principle, one can exhibit the higher-order crosscorrelation structure of the subthreshold voltage dynamics by simultaneously recording more than two neurons. However, these recordings are exceedingly challenging to perform and we are not aware of any reports of such measurements. That said, higher-order correlation statistics can be straightforwardly explored in single (or pair) recordings by investigating higher-order moments of the voltage traces. In particular, one can measure the asymmetry of the voltage distribution via its skewness, which is defined as the scaling-invariant quantity $\mathbb{S}[V] = M_{V,V,V}/M_{V,V}^{3/2}$, where $M_{V,V}$ and $M_{V,V,V}$ denotes the second and third centered moments of the stationary voltage $V$, respectively. *In-vivo* electrophysiological measurements have revealed that voltages are generally right-skewed, i.e., $\mathbb{S}[V] > 0$, with unimodal voltage distributions exhibiting comparatively heavier right tails than left tails [4,22]. Furthermore, it has been shown that this positive skewness varies with physiological synaptic drive. Specifically, spontaneous activity exhibits considerable positive skewness, with values as large as $\mathbb{S}[V] \sim 3$, whereas evoked activity displays significantly reduced skewness, with values as low as $\mathbb{S}[V] \sim 0.1 – 0.2$, compatible with a near-symmetric, Gaussian voltage distribution [4,22].

The skewness measurements discussed above are essentially independent from the spiking regime of the recorded neuron, which supports that voltage skewness originates from subthreshold neuronal mechanisms. Consistent with this observation, one can provide an elementary, heuristic explanation for the emergence and activity-modulation of the voltage skewness by recognizing the antagonistic role of excitatory and inhibitory driving forces [27–29]. During spontaneous activity, the voltage sits close to $V_i$ and far away from $V_e$ on average, leading to a vastly larger excitatory driving force than the inhibitory driving force. As a result, resting equilibrium must be achieved by balancing many small hyperpolarization events by comparatively fewer and larger depolarization events, leading to a substantial right skew. During the evoked regime, the voltage moves closer to $V_e$ and away from $V_i$, leading to more commensurate excitatory and inhibitory driving forces. Therefore, evoked equilibrium can now be achieved by balancing comparable hyperpolarization and depolarization events, leading to near symmetric voltage fluctuations. Albeit simple, this explanation involves

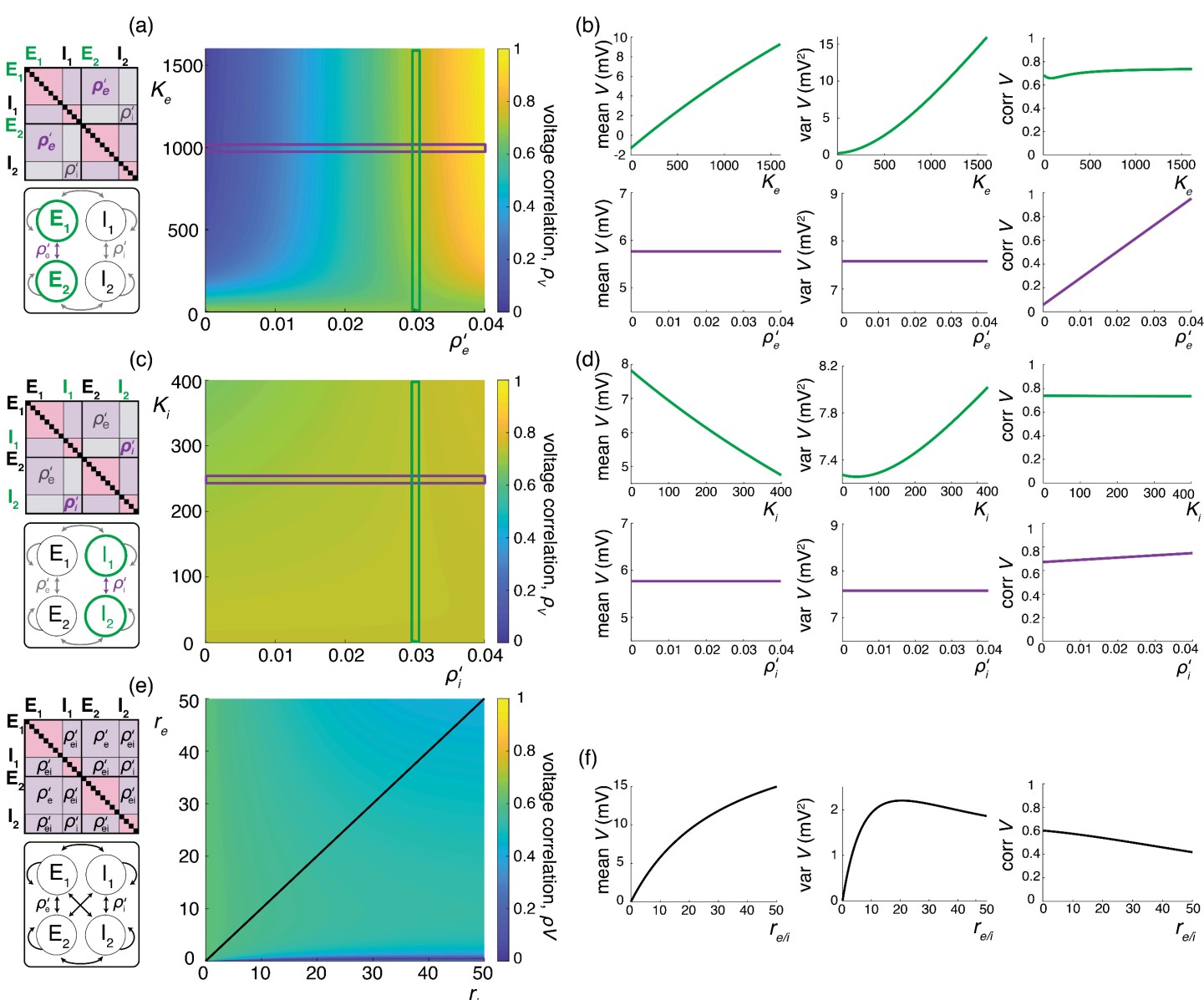

**Fig 7. Synaptic effects on voltage correlations through correlated input with moderately sized synaptic weights.** (a-b) Changes in neural statistics in response to a varying excitatory drive with $r_e = 10$ Hz and $\rho_e = 0.04$, while inhibition is held constant at $K_i = 250$, $r_i = 10$ Hz, $\rho_i = 0.04$, $\rho_i' = 0.03$, and $\rho_{ei} = \rho_{ei}' = 0$. (a) Voltage correlation as a function of excitatory inputs $K_e$ and cross excitatory correlation $\rho_e'$. (b) Changes in voltage statistics (mean, variance, and voltage correlation) while holding either a constant cross excitatory correlation of $\rho_e' = 0.03$ (top, green), or a constant $K_e$ of $10^4$ (bottom, purple). (c-d) Changes in neural statistics in response to a varying inhibitory drive with $r_i = 10$ Hz and $\rho_i = 0.04$, while excitation is held constant at $K_e = 10^4$, $r_e = 10$ Hz, $\rho_e = 0.04$, $\rho_i' = 0.03$, and $\rho_{ei} = \rho_{ei}' = 0$. (c) Voltage correlation as a function of inhibitory inputs $K_i$ and cross inhibitory correlations $\rho_i'$. (d) Changes in voltage statistics (mean, variance, and voltage correlation) while holding either a constant cross inhibitory correlation of $\rho_i' = 0.03$ (top, green), or a constant $K_i$ of 250 (bottom, purple). (e-f) Changes in neural statistics in response to increasing drive with $K_e = 10^4$, $K_i = 250$, $\rho_e = \rho_i = \rho_{ei} = 0.02$ and $\rho_e' = \rho_i' = \rho_{ei}' = 0.013$. (e) Voltage correlation as a function of input rate $r_e$ and $r_i$ in Hz. (f) Changes in voltage statistics (mean, variance, and voltage correlation) while jointly increasing both the excitatory and inhibitory drive $r_{e/i}$. In all cases the synaptic weights are moderately sized with $4W_e = W_i = 0.004$.

the tuning of many possible contributing factors, including connectivity numbers, synaptic weights, driving rates, but also input synchrony. In the following, we leverage our jump-process-based framework to show that explaining the observed skewness actually requires

an alternative explanation primarily involving the antagonistic role of relaxation and excitation alone. Furthermore, we also show that including physiological levels of input synchrony is necessary to quantitatively explain this skewness.

**Analytical predictions for excitation alone.** Applying the PASTA principle within our jump-process-based framework allows one to derive generic explicit formulas for any centered moments of an AONCB neuron's voltage. Although these formulas quickly become unwieldy for higher orders, one can express the third stationary, centered moment under the compact form

$$M_{V,V,V} = \frac{\mathbb{E}\left[(R-m)^3(1-Y)^3\right] + 3M_{V,V}\mathbb{E}\left[(R-m)\left(Y^2-1\right)(1-Y)\right]}{3/(b\tau) + \mathbb{E}\left[1-Y^3\right]}, \tag{26}$$

where $M_{V,V}$ denotes the stationary voltage variance (see Eq (16)) and where the auxiliary random variables $Y$ and $R$ are defined as

$$Y = e^{-(W_e + W_i)} \quad \text{with} \quad R = \frac{W_e V_e + W_i V_i}{W_e + W_i} - m.$$

As usual, the expectation in Eq (26) is with respect to the jump variable $(W_e, W_i)$, whose distribution parametrizes input synchrony. In the small-weight approximation, one can further exhibit the dependence of $M_{V,V,V}$ on the synchrony input structure in terms of the generalized spiking correlation coefficients defined in Eq (5). In general, this approach yields expressions for $M_{V,V,V}$ in terms of the second-order and the third-order spiking-correlation coefficients $\rho_{\alpha\beta,kl}$ and $\rho_{\alpha\beta\gamma,klm}$, $\alpha, \beta, \gamma \in \{e, i\}$. For simplicity, we only give this expression for the case of excitation alone with uniform synaptic weights and rates. For this case, we have

$$\frac{M_{V,V,V}}{(V_e - m)^3} = \frac{K_e r_e w_e^3\left((\rho_{e,3}(K_e - 2) + \rho_e)(K_e - 1) + 1\right)}{3(1/\tau + K_e r_e w_e)} - \frac{\left(K_e r_e w_e^2\left(1 + \rho_e(K_e - 1)\right)\right)^2}{(1/\tau + K_e r_e w_e)^2},$$

where $\rho_{e,3}$ denotes the uniform third-order spiking correlation coefficient. The above expression shows that even in the absence of inhibition, $M_{V,V,V}$ results from the contribution of two terms with opposite signs, suggesting that zero skewness may be achieved even without the antagonistic action of inhibition. To check this point, we adopt the beta-binomial statistical model from [6], which involves a single correlation coefficient via $\rho_{e,3} = 4\rho_e^2/(1 + \rho_e)$. Then, assuming that $K_e w_e \simeq 1$ as suggested by biophysical considerations, one has the approximate skewness expression

$$\mathbb{S}[V] \simeq \mathbb{s}[V]\left(\frac{1 - 2r_e\tau}{\sqrt{1 + r_e\tau}}\right)\sqrt{1 + \rho_e K_e} \quad \text{with} \quad \mathbb{s}[V] = \frac{2\sqrt{2}}{3\sqrt{K_e r_e \tau}}. \tag{27}$$

In the expression above, the quantity $\mathbb{s}[V]$ is the voltage skewness of the asynchronous, current-based model with identical parameters and driving inputs, which corresponds to considering a constant driving force $V_e$ instead $V_e - V$ in the dynamics given by Eq (1) without inhibition. Thus, even in the absence of a voltage-dependent driving force, the voltage is right-skewed in the spontaneous regime with, e.g., $\mathbb{s}[V] \simeq 0.25$ for $K_e = 10^4$ and $r_e \simeq 1$Hz, whereas this skew decreases with the input drive with, e.g., $\mathbb{s}[V] \simeq 0.05$ for $K_e = 10^4$ and $r_e \simeq 25$Hz. For large input numbers $K_e \geq 10^4$, the characteristic time between transient depolarization $1/(K_e r_e)$ remains smaller than the relaxation time $\tau$, even in the spontaneous regime for which $r_e \simeq 1$Hz. As a result, the voltage trace is nearly piecewise linear so that its

skewness is directly inherited from the Poissonian fluctuations of the input spike count over the timescale $\tau$. These become nearly Gaussian distributed—and thus unskewed—for large rate $K_e r_e \gg 1/\tau$, explaining the overall decreasing trend of the voltage skewness with input drive.

However, the right-skew exhibited by the asynchronous current-based model is generally too small in the spontaneous regime and too large in the driven regime compared to the values reported experimentally. To remedy these discrepancies, one can increase the spontaneous-regime skewness by including input synchrony. At the same time, one can decrease the driven-regime skewness by considering a voltage-dependent driving force, as in the AONCB models. Indeed, for all synchrony conditions, Eq (27) shows that the voltage of excitatory-driven AONCB neurons remains right-skewed in the driven regime up to $r_e \simeq 1/(2\tau) \simeq 33$Hz and maintains a small negative skewness for larger drive. The emergence of this small negative skew follows from the modulations of the Poissonian input fluctuations enacted by the voltage-dependent driving force, which amplifies downward input count fluctuations (larger driving forces) compared to upward ones (smaller driving forces). That said, this modulation is negligible at low driving rate so that increasing the magnitude of the skewness requires to act on the extra factor capturing the impact of synchrony. One can check that for $K_e = 10^4$ and $r_e = 1$Hz, we have $\mathbb{S}[V] \simeq 0.2$ for $\rho_e = 0$ but $\mathbb{S}[V] \simeq 1.3$ for $\rho_e = 0.03$. Altogether, the above discussion shows that realistic levels of voltage skewness can be explained by excitation alone in conductance-based neurons if these are driven by synchronous inputs.

**Numerical results for excitation and inhibition.**  A principled treatment of the general expression for the skewness of an AONCB neuron is burdensome (see Eq (26)) and perhaps more importantly, unnecessary to understand skewness under physiological conditions as we will see. For this reason, we resort to numerics to investigate skewness in the presence of excitatory and inhibitory driving inputs. We present our main results in Fig 8 where we show that for biophysically relevant parameters, the voltage skewness $\mathbb{S}[V]$ is almost exclusively shaped by excitatory inputs and that input synchrony remains a requirement to achieve realistic levels of skewness in the presence of excitation and inhibition. In Fig 8a we show two representative traces for the subthreshold membrane voltage of an AONCB neuron driven by asynchronous and synchronous inputs, as well as the corresponding voltage distributions. For moderate depolarization, the impact of synchrony is to substantially enhance excitation-driven depolarization, leading to rightly skewed distributions. In Fig 8b, we confirm that these asynchronous and synchronous skews can be accurately estimated via Eq (26). Specifically, we show that the PASTA analysis developed in S1 Text, S3 Appendix, produces accurate estimates of the higher-order voltage moments.

Next, we show that realistic levels of skewness, i.e., $\mathbb{S}[V] > 1$ in the spontaneous regime and $\mathbb{S}[V] < 0.1$ in the driven regime, can be achieved in the presence of both excitation and inhibition. In Fig 8c, we represent $\mathbb{S}[V]$ as a function of the driving rate $r$ in three driving conditions: excitation alone ($r_e = r$, $r_i = 0$), inhibition alone ($r_e = 0$, $r_i = r$), and joint excitation and inhibition ($r_e = r_i = r$). Consistent with intuition, at low input drive, the voltage fluctuations are dominated by transient hyperpolarizing or depolarizing inputs when driven by inhibition or excitation alone, respectively. This leads to right-skewed voltage distributions when $r_e > 0$, $r_i = 0$ and left-skewed distributions when $r_e = 0$, $r_i > 0$. Both skews vanish with increasing drive, as the mean voltage increases or decreases away from the resting potential but with values that remains away from the reversal potentials $V_e$ and $V_i$. In both cases, the same mechanism of inheritance of the Poissonian input fluctuations explains this reduction in skewness. However, considering excitation and inhibition together for physiological conditions reveals that the excitation-related mechanism is overwhelmingly responsible for the

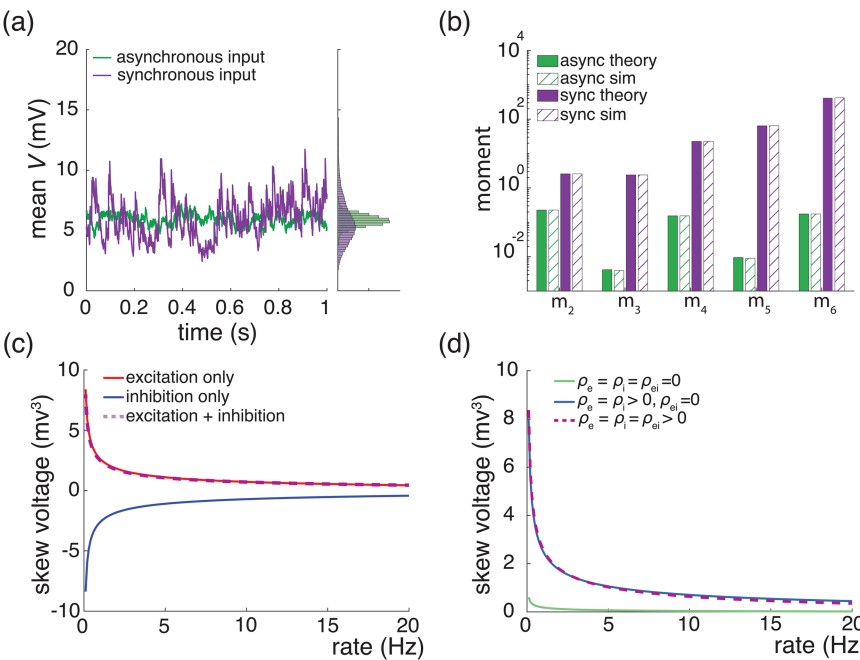

**Fig 8. Voltage is rightly skewed when presented with weakly-synchronized inputs at low drive.** (a) Example Monte-Carlo simulations of AONCB neurons, and voltage distributions, with asynchronous (green) and synchronous (purple) inputs. (b) Comparison of analytically derived moment expressions using equation (**??**) with numerical estimates obtained via Monte-Carlo simulations for the asynchronous (green) and synchronous (purple) conditions considered in (a). (c) Effects of increasing excitatory and/or inhibitory rate on skewness for excitatory inputs only (red), inhibitory inputs only (blue) or joint excitation and inhibition (purple). (d) Effects of jointly increasing excitatory and inhibitory input rate on skewness for various type of synchrony-based input correlations: uncorrelated $\rho_e = \rho_i = \rho_{ei} = 0$ (uncorr, green), within correlation $\rho_e, \rho_i > 0$ and $\rho_{ei} = 0$ (within corr, purple), within and across correlation $\rho_e, \rho_i, \rho_{ei} > 0$ (across corr, blue).

voltage skewness. This is due to the large difference between the magnitudes of the driving forces $|V_e - V| \gg |V_i - V|$, which gets nonlinearly magnified in the second and thrid moments involved in the skewness definition. For simplicity, we only consider within group synchrony in Fig 8c, which supports that voltage skewness does not arise from the antagonistic actions of excitation and inhibition but is almost entirely due to shot-noise excitatory drive. In Fig 8d, we show that including synchrony between excitation and inhibition does not alter skewness, consistent with the negligible role played by inhibition. Note that the predicted skewness range of values and trends are in agreement with observations reported *in vivo* when synchrony is included, whereas erasing synchrony altogether yields unrealistically low levels of skewness [4].

## Discussion

### Synchrony modeling and variability analysis

In this work, we have generalized the analytical approach proposed in [6] to quantify the impact of synaptic input synchrony on the membrane voltage variability of the so-called all-or-none-conductance-based (AONCB) neuronal model. Our generalization proceeds in two directions: First, we have extended our jump-process-based framework to model the synchronous drive to AONCB neurons resulting from inputs with heterogeneous rates and correlation structures. Such an extension allows for the consideration of more realistic input

models than the ones considered in [6], which relied on the restrictive hypothesis of input exchangeability. This involves the introduction of generalized higher-order spiking correlation coefficients which may vary according to the set of inputs considered. Second, we have adapted results from queueing theory to analyze the shot-noise-driven dynamics of AONCB neurons. These results hold in the limit of instantaneous synapses for which one assumes that the synaptic integration timescale is much smaller than the membrane time constant of the neuron. Such an approach allows one to extend the results of [6] to compute the arbitrary-order, centered, voltage moments of any feedforward assemblies of neurons driven by inputs with heterogeneous synchrony structure. The obtained formulas are derived recursively from the fixed-point relation that the moments obey in the stationary regime. These formulas involve expectations with respect to the distribution of the jumps of the driving process which model synchronous inputs. Prior to this work, physics-inspired techniques have been used to compute the time-dependent moments for the subthreshold dynamics of conductance-based neuronal models driven by shot noise [71,72]. However, these techniques were applied in the asynchronous regime and produced moment formulas in integral forms where parametric dependencies can be hard to capture. Notably, these techniques were recently expanded to include the impact of spiking activity on the time-dependent mean and auto-correlation structure of the neuronal response [73,74].

Our previous work supports that accounting for the surprising large variance of the membrane voltage measured *in vivo* necessitates weak but nonzero input synchrony, consistent with spiking correlation measurements in large spiking neuronal population [6]. Here, we have utilized our newly derived results to investigate whether weak but nonzero synchrony is consistent with other statistics of the measured subthreshold neuronal activity, including voltage covariability across pairs of neurons [17–21] and voltage skewness in single recordings [3,4,22]. We find that weak but nonzero input synchrony consistently explains the large degree of voltage correlations observed *in vivo* (0.4–0.8). This is by contrast with asynchronous inputs for which the emergence of similarly strong voltage correlations would imply that recorded neurons share at least 65% of their excitatory inputs, inconsistent with anatomical studies [66]. Moreover, we find that excitation is the primary driver of voltage covariability owing to the generally stronger driving force associated to excitatory synaptic events. Inspecting the detailed impact of the input correlations structure confirm this primary role of excitation as shown in Fig 9, which shows that excitation-specific correlations are the determinant of the voltage covariability. At the same time, while it plays a secondary role, inhibition can modulate voltage covariability in the evoked regime, when a relative increase in the inhibitory driving force partially cancels the excitation-induced correlations. However, such a modulation can also lead to an overall reduction in variability, akin to the variability quenching reported for non-primate mammals [1], but contrary to *in-vivo* observations in primates [4]. Similarly, we find that excitation is also the primary driver of the subthreshold voltage skewness measured *in vivo* and that explaining the modulation of skewness by the drive does not require inhibition to first approximation. However, we found that explaining the high degree of skewness observed in the spontaneous regime requires including weak but nonzero input synchrony, consistent with physiological observations in primates [4]. We devote the remaining of this work to discuss potential limitations to our analysis and future implications of our results.

### Interpretation in the small-weight approximation

A caveat of our exact moment calculations is that the obtained formulas can be difficult to interpret in relation to measurable quantities (see, e.g., Eq (16)). This difficulty stems from

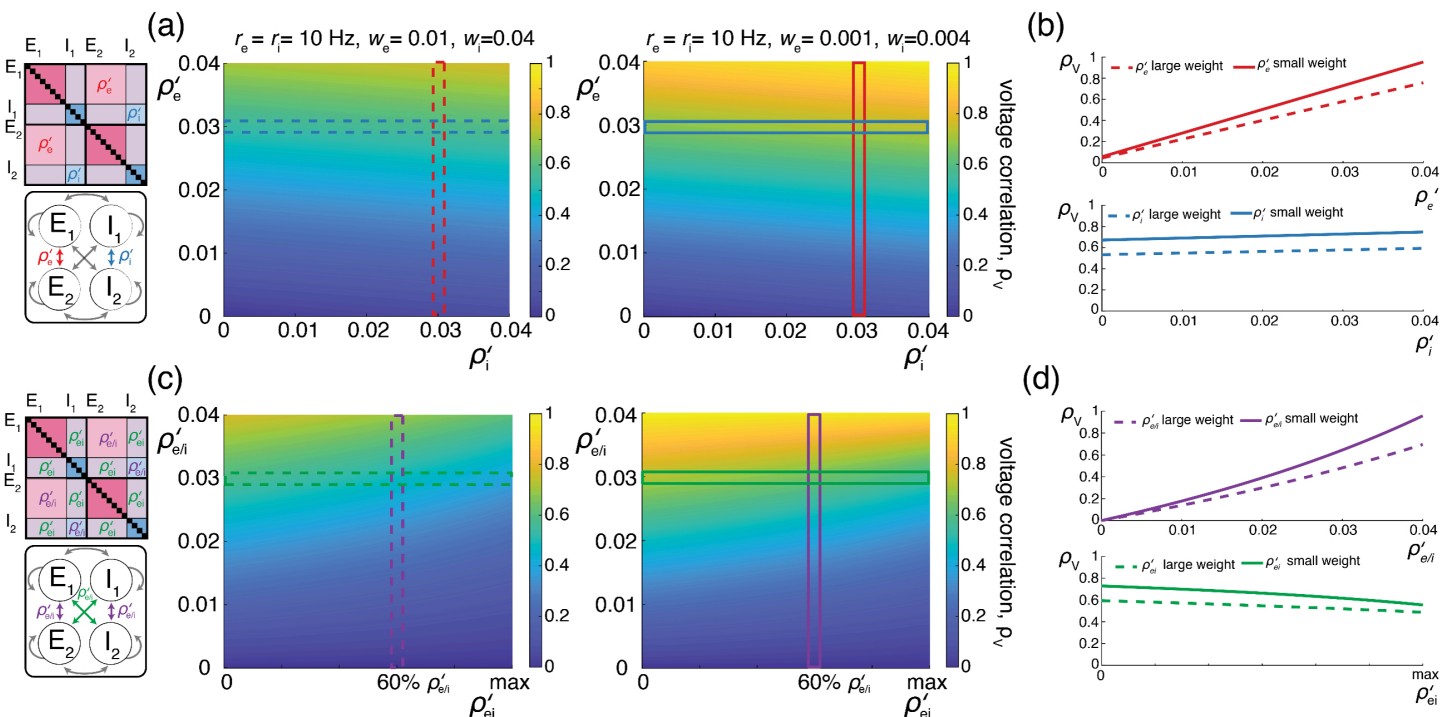

**Fig 9. Voltage correlations are driven primarily by excitatory-to-excitatory input correlations**. (a-b) Changes in cross-neuron voltage correlations in response to varying excitatory and inhibitory cross-neuron correlations $\rho'_e$ and $\rho'_i$, with $K_e = 10^4$, $K_i = 250$, $r_e = r_i = 10$ Hz, $\rho_e = \rho_i = 0.04$, and $\rho_{ei} = \rho'_{ei} = 0$. (a) Voltage correlations with large synaptic weights (left) and moderate synaptic weights (right). (b) Changes in voltage correlations while holding either a constant cross-neuron excitatory correlation of $\rho'_e = 0.03$ (top, red) or cross-neuron inhibitory correlation of $\rho'_i = 0.03$ (bottom, blue) for both large (dashed) and moderately sized (solid) synaptic weights. (c-d) Changes in cross-neuron voltage correlations in response to varying same-pool cross-neuron correlations $\rho'_e/\rho'_i$ and across-pool cross-neuron correlations $\rho'_{ei}$, with $K_e = 10^4$, $K_i = 250$, $r_e = r_i = 10$ Hz, $\rho_e = \rho_i = \rho_{ei} = 0.04$, and $\max(\rho'_{ei}) = \sqrt{\rho'_e \rho'_i}$. Changes in voltage correlations while holding either a constant same-pool cross-neuron correlation of $\rho'_e = \rho'_i = 0.03$ (top, purple) or cross-pool correlation of $\rho'_{ei} = 0.6\rho'_{e/i}$ (bottom, green) for both large (dashed) and moderately sized (solid) synaptic weights.

the fact that when the moments under consideration involve an assembly of (more than one) neurons, the corresponding formulas feature the rates of spiking events experienced by the assembly of neurons collectively (see S1 Text, S1 Appendix). In the presence of synchrony, these collective rates do not behave additively as several neurons can experience input activations at the exact same time. One can gain insight about these collective rates by considering a parametric model for input synchrony. For simplicity, let us consider the homogeneously synchronous beta-binomial model introduced in [6]. This model considers $K$ exchangeable inputs whose synchrony is parametrized by a single parameter $\beta > 0$ such that the pairwise spiking correlation is given by $\rho = 1/(1 + \beta)$. Given identical individual spiking rate $r$ and $\beta > 0$, the collective rate of spiking event experience by $K$ neurons can be shown to be

$$r_{1,\dots,K} = \left(\psi(K + \beta) - \psi(K)\right)\beta r \overset{K \to \infty}{\sim} (\ln K)\beta r,$$

where $\psi$ denotes the digamma function. Thus, in the presence of synchrony, the collective rate $r_{1,\dots,K}$ grows strictly sublinearly (logarithmically) as a function of the number of inputs. Such a logarithmic growth is characteristic of rich-gets-richer clustering processes, a view that can be adopted to describe the beta-binomial model for synchrony [75,76]. In this view, one builds synchronous input activity iteratively by specifying how to add a $(K + 1)$-th extra input

to an existing population of $K$ inputs while maintaining overall homogeneous synchrony. To do so over a period $T$, one allocates extra-input spikes to already existing spiking events—or clusters—of size $1 \leq k \leq K$ with probability $k/(\beta + K)$, while the extra input spikes on its own to start $P_\beta$ new clusters, where $P_\beta$ is a Poisson random variable with parameter $\beta T/(\beta + K)$. Thus, the number of new spiking events is inversely proportional to the number of inputs on average, leading to a logarithmic growth of their average total number of spiking events, and therefore of the collective rate $r_{1,...,K}$. We expect such logarithmic growth to be a generic feature of the collective drive exerted by synchronous inputs, but exactly computing such drives for heterogeneously synchronous inputs is generally a challenging task (see S1 Text, S6 Appendix).

Fortunately, the small-weight approximation allows one to sidestep the need to specify the collective rates of heterogeneous input assemblies, while making it possible to interpret our moment calculations in relation to directly measurable quantities. The small-weight approximation assumes that with strong probability, the total jump size remains small, i.e., $W_e + W_i \ll 1$. Such an assumption, which is justified by numerical simulations, is natural given the weak level of observed spiking correlations, i.e., $0.01 \leq \rho \leq 0.04$, and given the constraint that $K_e w_e = K_i w_i \simeq 1$, even for large connectivity numbers. Under this assumption, the small-weight approximations are obtained by Taylor expanding the exponential terms featured in our exact moment formulas. Importantly, the resulting approximate expressions can then be expressed in terms of individual input spiking rate $r_k$ (as opposed to the collective rates) and of the generalized correlation coefficients $\rho_{\alpha_1,...,\alpha_n,k_1,...,k_n}$. These coefficients defined in Eq (5) measured the propensity of the set of inputs $k_1, ..., k_n$ to activate simultaneously and is expected to decay with $n$, the number of considered inputs. For the beta-binomial model considered above with a single type of inputs, one can check that

$$\rho_{1,...,n} = \prod_{k=1}^{n-1} \frac{k}{\beta + k} = \frac{\Gamma(1 + \beta)\Gamma(n)}{\Gamma(n + \beta)} \overset{n \to \infty}{\sim} \frac{\Gamma(1 + \beta)}{n^\beta} \quad \text{with} \quad \beta = 1/\rho - 1,$$

where $\Gamma$ denotes the standard gamma function and where we recall that $\rho$ denotes the pairwise spiking correlation. Thus, $\rho_{1,...,K}$ vanishes as a power law with exponent $\beta = 1/\rho - 1$ when $K \to \infty$. This is consistent with intuition, which suggests that higher degree of synchrony, i.e., smaller $\beta > 0$, corresponds to slower decay of the high-order correlations.

## More realistic forms of synchrony

A possible limitation of our analysis stems from our modeling of spiking synchrony via jump processes, which enacts a perfect form of synchrony with exactly simultaneous synaptic activations. Considering instantaneous synchrony overlooks the fact that empirical spiking-correlation estimates such as

$$\rho_{i,j}(\Delta t) = \mathbb{C}_2 \left[ N_i(\Delta t), N_j(\Delta t) \right] / \sqrt{\mathbb{V} \left[ N_i(\Delta t) \right] \mathbb{V} \left[ N_j(\Delta t) \right]},$$

depend on the timescale $\Delta t$ over which the processes $N_i$ and $N_j$ count the spiking events of inputs $i$ and $j$. Experimental measurements have revealed that $\rho_{i,j}(\Delta t)$ decreases with smaller timescale $\Delta t$ and vanishes in the limit $\Delta t \to 0$ [7,8]. One can conveniently reproduce these experimental observations in more realistic models of synchrony obtained by jittering instantaneously synchronous spikes. This is because moving individual spiking times with independently and identically distributed time shifts, e.g., with centered Gaussian variables with standard deviation $\sigma_J$, erases temporal correlations at timescale shorter than $\Delta t < \sigma_J$.

Such an erasure implements the desired decrease of spiking correlations $\rho_{i,j}(\Delta t)$ with smaller timescale $\Delta t$, while causing an overall decrease of $\rho_{i,j}(\Delta t)$ over all timescales $\Delta t$ compared with the instantaneous, unjittered spiking correlations. In fact, one can produce empirical correlations with biophysically relevant temporal profile by jittering instantaneously synchronous spikes with $\rho_e = 0.25$ with $\sigma_J = 50$ms.

It turns out that considering the above, more realistic forms of jittered synchrony reveals that although unrealistic, perfect input synchrony still yields biologically relevant estimates of the stationary voltage statistics. Indeed, as noticed in [6], we find that the stationary voltage statistics of AONCB neurons, even for realistic forms of synchrony, are essentially determined by the input synchrony measured over a single time scale. We find this relevant timescale to be around $\Delta t \simeq 25$ms, about twice the duration of the membrane time constant over which inputs are integrated by the neurons. Accordingly, the biophysically relevant range of spiking correlations that we consider in this work, i.e., $\rho_e \simeq 0.01 - 0.04$, corresponds to measurements performed over the timescale $\Delta t \simeq 25$ms [7,8]. To confirm these observations, we validate our synchrony modeling approach in Fig 10, where we compare the response of the same AONCB neurons to instantaneously synchronous inputs and to jittered synchronous inputs. Specifically, in Fig 10a, we show representative voltage responses for an AONCB neurons driven by matched unjittered and jittered synchronous inputs. Although these traces exhibit distinct temporal dynamics, Fig 10b reveals that both forms of synchronous inputs yield nearly identical stationary voltage distributions, and thus nearly identical skewness coefficients as well. Similarly, in Fig 10c, we show representative voltage responses for a pair of AONCB neurons driven by matched unjittered and jittered synchronous inputs. We then check in Fig 10d, that these temporally distinct dynamics yield nearly identical stationary joint voltage distributions, and thus nearly identical voltage correlations as well. Thus, although our approach produces distinct transient responses depending on the form of the synchronous drives, it yields consistent stationary statistics.

## Obscuring role of synaptic heterogeneity

As noted when introducing biophysical parameters, an apparent limitation of our analysis is the consideration of typical synaptic weights. This assumption implies that conductance variability exclusively stems from the variable numbers of coactivating synapses, and thus from synchrony. This is certainly an oversimplification as synapses exhibit heterogeneity in weights [55], which must play a role in shaping neural variability [54]. Distinguishing between heterogeneity and synchrony contributions, however, is a fundamentally ambiguous task [77]. For instance, consider $K$ inputs with identical weight $w$ and identical spiking rate $r$. Such inputs are instantaneously synchronous if $p(k)$, the probability that $k$ of them coactivate, is not concentrated on $k = 1$ but also distributed on values $k > 1$. It is not too hard to realize that such a synchronous model is indistinguishable from considering $K$ independent inputs with heterogeneous weights $kw$ and associated activation rates $Krp(k)$, $1 \leq k \leq K$. This ambiguity mirrors that of interpreting whether an EPSP results from a single or multiple coincidental synaptic activations. That said, recent experimental evidence supports that cortical response selectivity derives from strength in numbers of synapses, rather than difference in synaptic weights [78], consistent with a dominant role for synchrony.

In any case, our framework is general enough to account for the impact of heterogeneities. Actually, all our analytical results are derived assuming synaptic weights heterogeneities and these can be used to gauge the respective contribution of synaptic heterogeneity and input synchrony to neuronal variability. Performing such an analysis reveals that in the absence of synchrony, the heterogeneity contribution is primarily controlled by the coefficient of

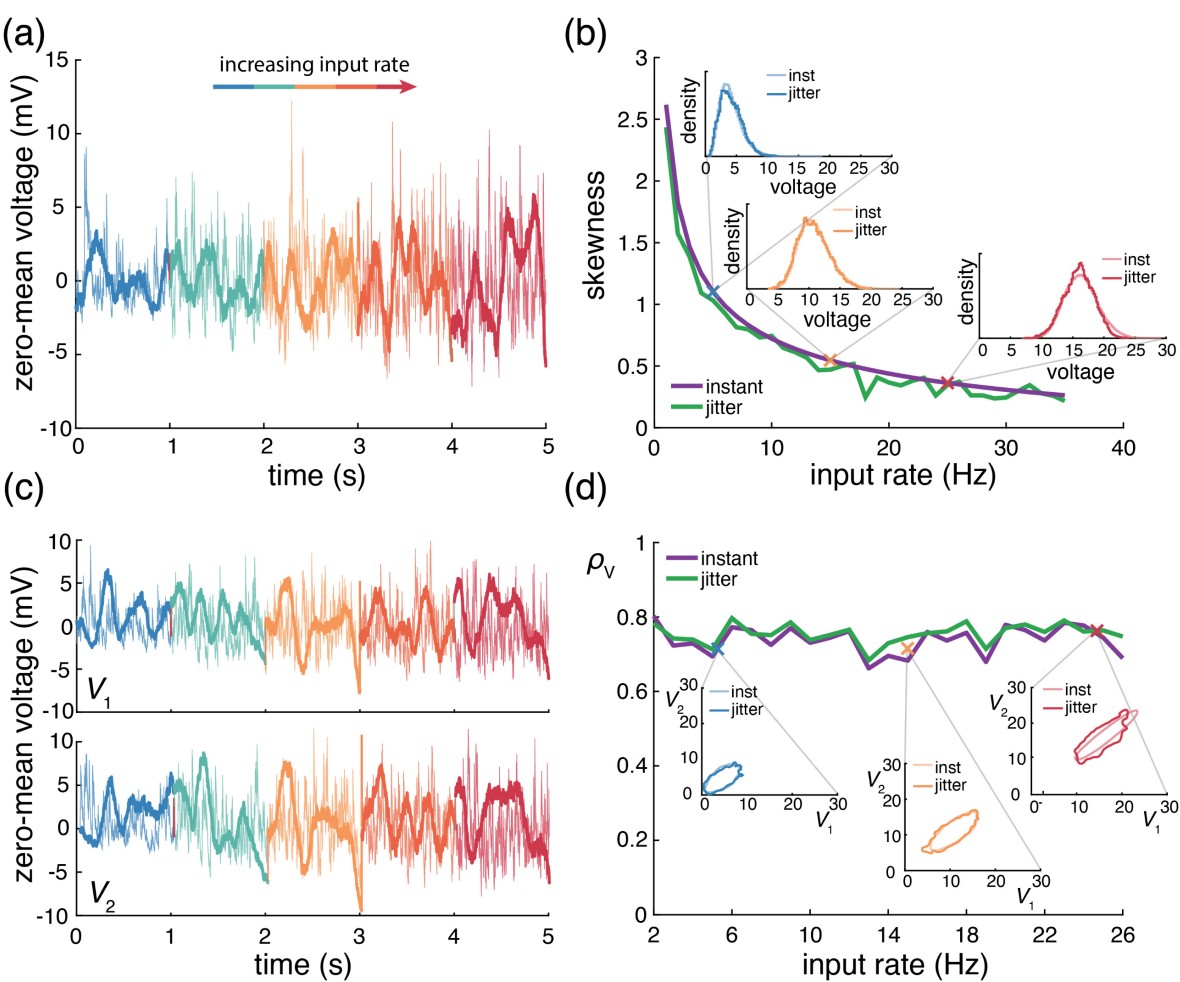

**Fig 10. Theory is consistent when introducing jitter into synchronized inputs.** (a) Example Monte Carlo simulations of mean-subtracted AONCB neurons under both instantaneous synchrony (thin line) and jittered input synchrony (thick line), with increasing excitatory input rates (warming colors). (b) Changes in skewness with increasing excitatory input rates for both instantaneous synchrony (purple) and jittered synchrony (green). Insets show voltage histograms corresponding to simulations in (a), color-coded to match. (c) Example Monte Carlo simulations of mean-subtracted AONCB neurons for two neurons ($V_1$, top and $V_2$, bottom), with cross-neuron input correlations $\rho'_e$ under both instantaneous synchrony (thin line) and jittered input synchrony (thick line), with increasing excitatory input rates (warming colors). (d) Changes in voltage correlations with increasing excitatory input rates for both instantaneous synchrony (purple) and jittered synchrony (green). All simulations use $K_e = 10^4$ with moderate synaptic weight sizes. For instantaneous input synchrony, the input correlation is $\rho_e = 0.03$, while for jittered input synchrony, the instantaneous correlation is $\rho_e = 0.25$ with a spike jitter of 50 ms, resulting in a spiking correlation of 0.03 when observed within a 25 ms window.

variation of the synaptic weights, denoted by CV[$w$]. To see why, let us consider the case of asynchronous heterogeneous excitatory inputs but with fixed rate $r$ for simplicity. In the small-weight approximation, the subthreshold variance (21) simplifies to:

$$M_{V,V} = \frac{r \sum_{k=1}^{K} w_k^2 (V_e - m)^2}{2(1/\tau + r \sum_{i=1}^{K} w_k)}.$$

Identifying the typical value of the synaptic weights to the mean synaptic weight $\langle w \rangle$, we then have:

$$\sum_{k=1}^{K} w_k^2 = K\langle w^2 \rangle = K\left(\langle w \rangle^2 + \langle w^2 \rangle - \langle w \rangle^2\right) = K\langle w \rangle^2(1 + \mathrm{CV}^2[w]).$$

where the angled brackets denote averages over synapses. This shows that the contribution of heterogeneity to variability is mediated by the squared coefficient of variation of the weights

$$M_{V,V} = \frac{rK\langle w \rangle^2(1 + \mathrm{CV}^2[w])(V_e - m)^2}{2(1/\tau + r\sum_{i=1}^{K} w_k)},$$

to be compared with the corresponding variance for the homogeneous, synchronous case:

$$M_{V,V} = \frac{rK\langle w \rangle^2(1 + \rho(K-1))(V_e - m)^2}{2(1/\tau + r\sum_{i=1}^{K} w_k)}.$$

Note that in the equations above, the term $\mathrm{CV}^2[w]$ formally plays the same role as the term $\rho(K-1)$, where $\rho$ denotes the input spiking correlation.

The coefficient $\mathrm{CV}[w]$ has been inferred from EPSP amplitude measurements in several experimental studies including [56,57]. These measurements indicate that $\mathrm{CV}[w]$ is smaller than 1 (about 0.4 in most studies). This supports that synaptic weight heterogeneity can at most double subthreshold variability, which is not enough to achieve realistic voltage variance with moderate synaptic weights. This is by contrast with the synchrony contribution [6]. Indeed, considering $10^3$ inputs with weak spiking correlation $\rho = 0.01$ (lower end of the measurement range) already yields $\rho(K-1) \simeq 10$, which corresponds to increasing variability by an order of magnitude. This supports our hypothesis that synchrony is the main driver of variability, at least compared with the contribution of synaptic heterogeneity.

### Effect of faulty synaptic transmission

Another possible limitation of our approach is that it neglects synaptic faultiness as a source of variability. Synaptic faultiness refers to the observation that chemical synapses release neurotransmitters stochastically in response to action potentials, so that the response of a postsynaptic neuron to a presynaptic spiking event may vary or fail altogether [79,80]. Because it amounts to considering additional sources of independent noise, we expect that including faulty synaptic transmission in our modeling framework will yield a decrease in the impact of synchrony, potentially challenging our results. To explore the impact of faulty synapses, let us model faulty synaptic activation variables as $X'_{\alpha,k} = B_{\alpha,k}X_{\alpha,k}$, where $B_{\alpha,k}$ denotes independent $\{0,1\}$-valued Bernoulli variables with parameters $\mathbb{E}[B_{\alpha,k}] = p_{\alpha,k}$. This corresponds to considering the case of all-or-none faultiness for which synaptic transmission fails with probability $1 - p_{\alpha,k}$. In this setting, and consistent with intuition, faulty transmission causes the effective synaptic rates and the effective spiking correlation coefficients to decrease according to

$$r'_{\alpha,k} = r_{\alpha_k}\mathbb{E}[B_{\alpha,k}] = p_{\alpha,k}r_{\alpha_k} \quad \text{and} \quad \rho'_{\alpha\beta,kl} = \frac{\mathbb{E}[X'_{\alpha,k}X'_{\beta,l}]}{\sqrt{\mathbb{E}[X'_{\alpha,k}]\mathbb{E}[X'_{\beta,l}]}} = \rho_{\alpha\beta,kl}\sqrt{p_{\alpha,k}p_{\beta,l}}.$$

In order to maintain a realistic mean voltage response range, one must compensate for the effective reduction in driving rates caused by synaptic faultiness. In principle, such compensation can be achieved by inversely scaling the synaptic weights according to $w'_{\alpha,k} = w_{\alpha,k}/p_{\alpha,k}$. However, such a scaling requires to introduce unrealistically large synaptic weights for high degrees of synaptic failure ($p_{kl} \simeq 0.1$) [50,51]. It is thus more natural to compensate for the effect of synaptic failure by scaling the number of synaptic inputs instead. Assuming homogeneous weights $w_\alpha$ and failure rates $1 - p_\alpha$, this amounts to set $K'_\alpha = K_\alpha/p_\alpha$, while holding $w'_\alpha = w_\alpha$. On can then check using formulas Eqs (23) and (27) that such a scaling of the input numbers also preserves the second- and third-order statistics of the voltage response in the small-weight approximation. Therefore, and perhaps counterintuitively, our results hold in the presence of faulty synaptic transmission at the mere cost of increasing the number of synaptic inputs. Such a cost is inconsequential as assuming drastic faultiness such as $p = 0.1$ involves considering inputs numbers $K_e, K_i \leq 10^4$, which remain compatible with anatomical observations [66].

## Effect of including gap junctions

One more limitation of our approach is that we have not investigated how gap junctions may impact subthreshold variability. Gap junctions establish instantaneous, bidirectional, electrical couplings between neighboring cells [81]. In cortex, these gap junctions are a distinct feature of inhibitory spiking cells, which form connected networks of electrically coupled cells [82,83]. Our framework can be extended to study the impact of such electrical couplings, which must clearly promote positive correlations across cells. For simplicity, consider an electrically coupled pair of neurons that only receive excitatory inputs. The voltages $(V_1, V_2)$ of the pair of neurons obey the following system of first-order stochastic equations

$$
\begin{aligned}
C\dot{V}_1 &= G(V_L - V_1) + G_{e,1}(V_e - V_2) + G_{12}(V_2 - V_1), \\
C\dot{V}_2 &= G(V_L - V_2) + G_{e,2}(V_e - V_2) + G_{12}(V_1 - V_2),
\end{aligned}
\tag{28}
$$

where $G_{12}$ denotes the electrical coupling induced by gap junctions between neuron 1 and 2. Because the system of Eq (28) is linear, applying the PASTA principle in the stationary regime still yields analytically tractable conservation equations. The main complication stems from the fact that in between consecutive spiking arrivals, the deterministic evolution of $V_1$ and $V_2$ is coupled via the constant term $g = G_{12}/G$. By contrast, one can check that the Marcus update rules Eq (10) still apply unchanged.

Although our analysis extend straightforwardly to include gap junctions, obtaining compact interpretable expressions for the resulting moments is a practical challenge. For conciseness, we only give here formulas for the first moment of the neuronal voltages. These can be shown to be weighted averages of the uncoupled means $m_1|_{g=0}$ and $m_2|_{g=0}$:

$$
m_1 = \frac{c_{11,g} m_1|_{g=0} + c_{12,g} m_2|_{g=0}}{c_{11,g} + c_{12,g}} \quad \text{and} \quad m_2 = \frac{c_{21,g} m_1|_{g=0} + c_{22,g} m_2|_{g=0}}{c_{21,g} + c_{22,g}},
\tag{29}
$$

where we have defined the averaging weights

$$
c_{ab,g} = (1/\tau + c_a)(1/\tau + c_b)\delta_{ab} + g(1/\tau + c_b), \quad a, b \in \{1, 2\}.
$$

In the above definition, the rate coefficients $c_a$, $a \in \{1, 2\}$, are such that $c_a = c_{e,a} + c_{i,a}$, where the coefficients $c_{\alpha,a}$, $\alpha \in \{e, i\}$, are those defined by (15) in the absence of gap junctions.

As intuition suggests, Eq (29) shows that the effect of electrical coupling between neurons 1 and 2 is to bring their voltages together, and in the limit of large coupling $g \to \infty$, we asymptotically have $m_1 = m_2$. Perhaps also not surprisingly, tedious but straightforward calculations show that electrical coupling between symmetrical cells increases voltage correlations across neurons, while marginally decreasing the voltage variance by averaging the impact of synaptic inputs across neurons. It would be interesting to generalize such calculations to higher order in order to better understand how electrically coupled network of cells may play a special role in promoting synchrony.

### Neglect of the spike-generating mechanism

Yet another limitation of our approach is that mathematical tractability was obtained at the cost of neglecting the impact of the cell's own spiking on its membrane voltage fluctuations. This neglect, however, is justified given the specific goal of our approach, which is to quantify the impact of input synchrony on the subthreshold variability observed *in vivo*. Consistent with this approach, the estimates of voltage variance, skewness, and covariance that we used (some of which were also used in [6]) were measured for voltage traces that exclude action potentials in primates [3,4] or were collected in the rodent somatosensory or visual cortex, where activity is typically low [18–21]. In other words, the experimental measurements that we use for comparison in our analysis are unlikely to be influenced by the cell's own spiking. It is also worth noting that the authors of [17] specifically tested whether the voltage correlations observed in the visual cortex of cats can be attributed to spiking in synchrony and concluded that these correlations were independent of the occurrence of action potentials. This is because *in vivo* subthreshold fluctuations in neurons whose spiking is precluded by hyperpolarizing current injections are similar to that of of spiking cells [84].

That said, we expect the cell's own spiking to impact membrane fluctuations in high-activity areas, even if one only considers voltage traces that exclude action potentials. In principle, this impact can be studied by supplementing AONCB neurons with an integrate-and-fire spiking mechanism. A PASTA treatment is still possible in this augmented setting but it only yields a fixed-point characterization of the moments under the form of integral equations that need to be solved numerically. The study of these equations is beyond the scope of this work. Preliminary simulations indicate that the inclusion of a hard reset close to the leak reversal potential is problematic when modeling subthreshold variability. Indeed, by causing neurons to reset far away from their spiking threshold, such a reset rule generally yields unrealistic negative skewness in the driven regime. By contrast, we empirically find that one can achieve realistic nonnegative skewness by implementing a reset to their stationary mean drive, which depends on the driving condition. The effect of such a rule is to attenuate the impact of the cell's own spiking on membrane voltage fluctuations. It is also consistent with the experimental observation that neurons reset near their mean voltage trend *in vivo*.

### Biophysical relevance and modeling implications

Spike-count correlations have been experimentally shown to be weak in cortical circuits [9,14,42,85]. For this reason, a putative role for synchrony in neural computations remains a matter of debate [86–88] while most theoretical approaches have been developed in the asynchronous regime of activity [34–40]. However, when distributed over large networks, weak spiking correlations can still give rise to impactful synchrony, once information is pooled from a large enough number of synaptic inputs [15,16]. After identifying input synchrony as the primary driver of subthreshold variability in [6], we have argued in this work that a

weakly synchronous regime of activity is also compatible with the large degree of voltage correlations observed in *in-vivo* pair recording. At one extreme, assuming that voltage correlations emerge from shared asynchronous inputs implies that neurons share an unrealistically large fraction of inputs. At the other extreme, weakly synchronous inputs can account for biophysically relevant voltage correlations, even without assuming any shared inputs. The latter explanation is compatible with the view that a strong, synchronous afferent drive dominating weak, recurrent, network interactions is the key to achieve realistic regimes of activity in conductance-based neurons [52,89]. That view also suggests an alternative explanation for the decline of voltage correlations in the evoked regime. Our analysis suggests that such a decline directly follows from the increase in the inhibitory driving-force induced by gradual depolarization, in agreement with the proposal that inhibition controls the regime of correlations in cortex [90]. However, in the presence of synchrony, this mechanism can lead to a form of variability quenching, which has been widely reported [1] but is inconsistent with *in-vivo* measurements in primates [4]. Alternatively, under the assumption that the synchrony regime is governed by the external drive, one can jointly achieve a reduction in voltage correlations and an increase in voltage variability by merely assuming that the external drive is less synchronous than the spontaneously synchronous activity. As illustrated in Fig 11, our analytical framework can still apply to this case and produce the desired neuronal response.

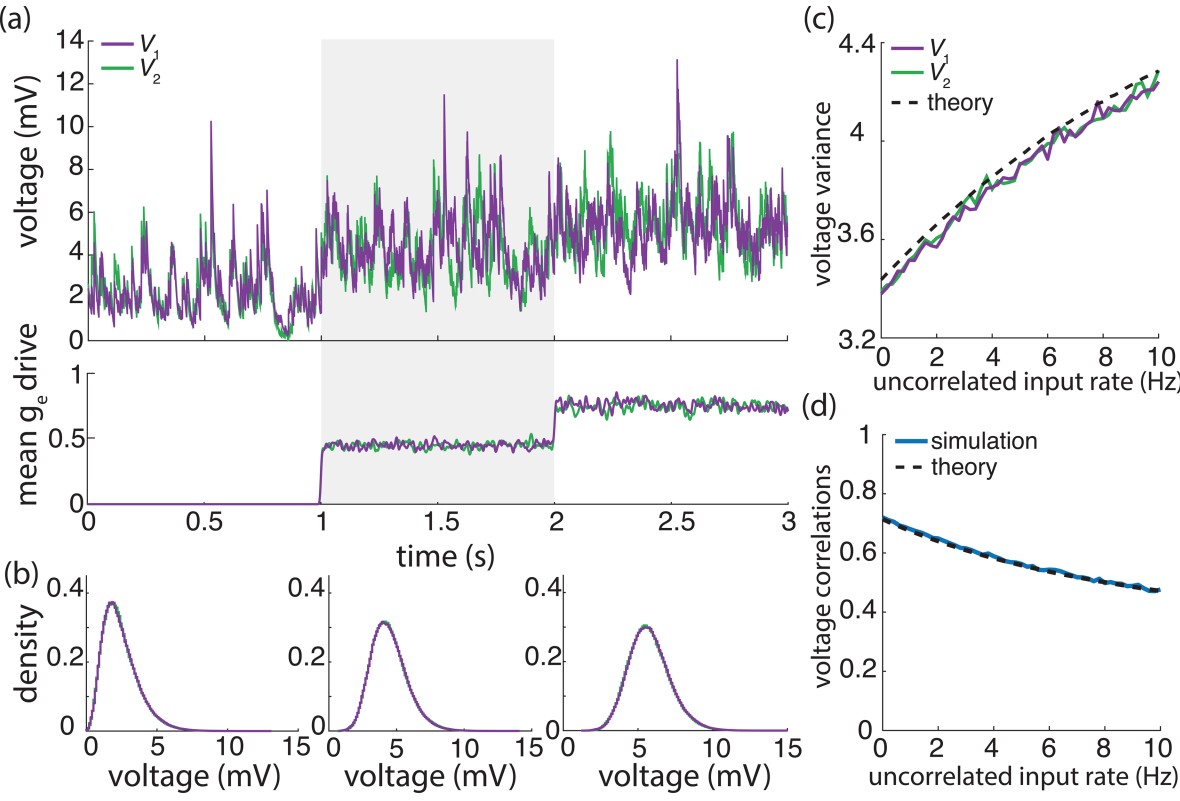

**Fig 11. External uncorrelated excitatory drive de-correlates paired neurons** (a) Example Monte Carlo simulations of 2 AONCB neurons (top) with increasing uncorrelated excitatory drive (bottom). Each neuron is driven by inputs with within-neuron correlations $\rho_e = \rho_i = 0.03$ and $\rho_{ei} = 0$ and cross-neuron correlations $\rho'_{e/i} = 0.025$ and $\rho'_{ei} = 0$. (b) Voltage histograms corresponding to the three levels of uncorrelated input drive shown in (a). (c) Change in voltage variance as uncorrelated input drive rate increases. (d) Same as (c) but for voltage correlations.

In this case, two neurons receive both within- and cross-neuron correlated inputs, characterized by $K_e = 10^4$, $K_i = 250$, $r_e = 5$ Hz, $r_i = 10$ Hz, $w_e = 0.001$, $w_i = 0.004$, $\rho_e = 0.03$, $\rho_i = 0.03$, and $\rho'_e = \rho'_i = 0.025$. Additionally, each neuron is driven by large, asynchronous excitatory inputs with $K_e = 250$ and $w_e = 0.01$, and varying rates, representing thalamic inputs driven by external stimuli.

That said, recurrent connections are likely contributing to shaping voltage synchrony, at least during spontaneous activity when external drive is absent [13,35,91,92]. Addressing this question requires to quantitatively understand how activity synchrony emerges in neural networks with prescribed degree of shared inputs. This involves a careful correlation analysis of large-but-finite network of spiking conductance-based neurons beyond classical mean-field techniques [52,53]. To date, most of these mean-field approaches have been conducted in balanced networks, whereby recurrent inhibition promotes an asynchronous regime of activity by counteracting correlation-inducing excitation [14]. Although a tight balance between excitation and inhibition implies asynchronous activity, [12,13,93], a loosely balance regime is compatible with the establishment of strong neuronal correlations [94–96]. However, a theoretical analysis of the emergence of these correlations is still lacking. In our previous work [6], we argued that such an analysis is hindered by the fact that correlations are finite-size phenomenon that cannot be studied via scaling limits in conductance-based neurons, where in particular, we show that the balanced scaling limit for which $w_e, w_i \sim 1/\sqrt{K}$, $K(w_e - w_i) = O(1)$ with $K \to \infty$ is only variability-preserving for current-based models. Another hurdle to characterize the emergence of network synchrony is that spiking correlations exhibit characteristic timescales that may vary according to regimes of activity [97–99]. Because it formally relies on an instantaneous model of synchrony, our tractable framework cannot account for the emergence and modulation of these characteristic timescales. There can be several biological origins for these characteristic timescales: they may represent the typical dwelling time of metastable network dynamics (e.g., up-and-down state dynamics), they may be a result of chemical neuromodulatory mechanisms [100,101], or they may be inherited from complex brain rhythms patterns or from other brain regions' top-down influences [102,103]. All these origins suggest analyzing the emergence of synchrony beyond our proposed framework in a multiscale dynamical setting.

## Supporting information

**S1 Text. S1–S6 Appendices.**
(PDF)

## Acknowledgments

We would like to thank Nicholas Priebe, David Hansel, and Eyal Seidemann for insightful discussions.

## Author contributions

**Conceptualization:** Logan A. Becker, François Baccelli, Thibaud Taillefumier.

**Formal analysis:** Logan A. Becker, François Baccelli, Thibaud Taillefumier.

**Funding acquisition:** François Baccelli, Thibaud Taillefumier.

**Investigation:** Logan A. Becker, Thibaud Taillefumier.

**Methodology:** Logan A. Becker, Thibaud Taillefumier.

**Project administration:** Thibaud Taillefumier.

**Resources:** Thibaud Taillefumier.

**Software:** Logan A. Becker, Thibaud Taillefumier.

**Supervision:** Thibaud Taillefumier.

**Validation:** Thibaud Taillefumier.

**Visualization:** Logan A. Becker, Thibaud Taillefumier.

**Writing – original draft:** Logan A. Becker, Thibaud Taillefumier.

**Writing – review & editing:** Thibaud Taillefumier.

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
