## [Decision Letter · Decision Letter 0]

22 May 2025

PCOMPBIOL-D-25-00525

Subthreshold moment analysis of neuronal populations driven by synchronous synaptic inputs

PLOS Computational Biology

Dear Dr. Taillefumier,

Thank you for submitting your manuscript to PLOS Computational Biology. After careful consideration, we feel that it has merit but does not fully meet PLOS Computational Biology's publication criteria as it currently stands. Therefore, we invite you to submit a revised version of the manuscript that addresses the points raised during the review process.

Please submit your revised manuscript within 60 days Jul 22 2025 11:59PM. If you will need more time than this to complete your revisions, please reply to this message or contact the journal office at ploscompbiol@plos.org. Please include the following items when submitting your revised manuscript:

We look forward to receiving your revised manuscript.

Kind regards,

Renaud Blaise Jolivet, Ph.D.

Academic Editor

PLOS Computational Biology

Marieke van Vugt

Section Editor

PLOS Computational Biology

**Additional Editor Comments :**

The reviewers have appreciated your manuscript entitled "Subthreshold moment analysis of neuronal populations driven by synchronous synaptic inputs," but have raised a number of concerns that you should thoroughly address in your revised manuscript. Note that I am not adamant on a detailed comparison to experiments (point raised by reviewer 3), but would of course appreciate it if that was possible.

**Journal Requirements:**

At this stage, the following Authors/Authors require contributions: Thibaud O. Taillefumier. Please ensure that the full contributions of each author are acknowledged in the "Add/Edit/Remove Authors" section of our submission form.

4) We notice that your supplementary information is included in the manuscript file. Please remove them and upload them with the file type 'Supporting Information'. Please ensure that each Supporting Information file has a legend listed in the manuscript after the references list.

2) If any authors received a salary from any of your funders, please state which authors and which funders.

**Reviewers' comments:**

Reviewer's Responses to Questions

Reviewer #1: This paper provides an analytically tractable approach aimed at expalining how neural synchrony affects subthreshold voltage variability in conductance-based neurons. The authors use a jump-process framework and derive exact expressions for voltage statistics. They demonstrate that weak but nonzero synchrony can explain experimentally observed voltage covariance between neurons and voltage skewness. They claim that asynchronous inputs cannot capture these statistics (although see comment below), and support the conclusions of [6] which challenge the prevailing asynchronous state hypothesis and suggesting that synchrony is a primary driver of cortical variability.

The paper is well written, and the number of results justify the its length. I appreciated the way the model was developed which made it possible to use biophysically realistic parameters (with exceptions), but also keep it analytically tractable. This provides a principled approach that is likely to be extendable and thus provide further insights into neuronal population dynamics. Although this is well trodden ground with analytical results that go back into the 1960s at least, the results and approach are novel, and the paper deserves to be published.

I do have several comments that should be addressed before publication: One of the main conclusions (as in their previous paper) is that if that only a small number of asynchronous input can explain the variability in membrane dynamics observed in vivo. The authors therefore claim that with physiologically realistic synapse counts synchronous inputs are more consistent with the data. However, synaptic weights vary greatly in size, and are approximately lognormally distributed in pyramidal cells in cortex (and elsewhere). Not taking into account that synaptic weights are broadly distributed limits the conclusions that can be drawn from the model. This should be address up front. I think that the approach is general enough to allow for such distributions.

Physiological spike trains often exhibit structured temporal correlations (bursting, refractoriness), which could influence neuronal variability. The authors note this, but it would be good to note that this temporal variability can also impact the results. Also, if synchronous spikes are not precisely aligned in time, I would assume that stronger correlations are needed to recover the same results. On the other hand, the temporal dynamics of synaptic inputs will affect the conclusions as well.

The authors also do not model negative correlations which could significantly impact dynamics. What would be the effect of negative correlations which are also observed often.

There are many other ways to model synchronous inputs - for instance see the work of Krumin and Shoham, Kuhnn, Aertsen and Rotter, and Trousdale, Shea-Brown and Josic. These approaches also allow for jittering. It would be good to discuss briefly how these previous models are related to the present model - and in particular whether approaches similar to those proposed in these previous papers could be used to get models where higher order moments can be better controlled and their impact analyzed. In particular, I assume that an approach similar to that used to define an MIP and SIP process could be used here to control the structure of higher order correlated variability across the population.

Smaller comments:

- Check for small grammatical errors, eg

l. 218 amounts to know -> amounts to knowing

l. 256 purpose -> purposes

and amusing typo on l. 312

- Define the acronym PASTA, unless I missed it

Additional note:

I did not check all the derivations, although the results seem plausible. It would be good

if the most important analytical expressions are checked against simulations. This is not

necessary for all expressions.

Reviewer #2: In the submitted manuscript "Subthreshold moment analysis of neuronal populations driven by synchronous synaptic inputs", the authors present a mathematical analysis of how the covariance and skewness of subthreshold activity cortical neurons is affected by synchrony in the spiking activity of the presynaptic neurons. The covariance and skewness of membrane potential fluctuations within All-Or-Nothing-Conductance-Based (AONCB) model neurons are derived analytically in a biologically plausible limit from an explicit set of equations for all statistical moments that is derived using the Poisson Arrivals See Time Averages (PASTA) principle from queuing theory. It is shown that experimentally observed variance, covariance, and skewness are all simultaneously predicted in a biologically realistic network only when observed levels of input synchrony are included. These analytical results are backed up and potential concerns regarding the unrealisticness of AONCB neurons are satisfactorily addressed through the use of numerical simulations.

Existing literature has demonstated that synchrony accounts for the variance in subthreshold and spiking activity in cortical neurons. This paper expands on and strengthens the conclusions of this literature by demonstrating that synchrony allow explains the subthreshold covariance between neurons and the subthreshold skewness in individual neurons. It also develops and utilizes innovative analytical techniques for studying subthreshold activity. The work presented in this manuscript is innovative, impactful, and technically sound; however, there are some issues with the presentation that warrant revision.

Most of the figures in the paper present results of Monte Carlo simulations, however some key details of the implementation of these simulations are not clearly specified. While the effect of input events on the membrane potential is thoroughly explained (i.e. the Marcus dynamics), it is unclear how input events with desired statistics are generated. In particular, the authors must include a clear description of how the $\left(\{X_{e,k}\}_{1\leq k\leq K_e},\, \{X_{i,l}\}_{1\leq l\leq K_i} \right)$ are chosen for each event time, in order to ensure the correct statistics for the inputs. Moreover, for the simulation results shown in Figure 11, it is unclear how the external uncorrelated excitatory drive is generated and included in the simulation, especially since the parameters for the simulation are given as $\rho_e=\rho_i=0.03$, $\rho_{ei}=0$, $\rho'_{e/i}=0.025$, and $\rho'_{ei}=0$. Do these parameters specify the background with the uncorrelated drive coming from a separate population of completely uncorrelated excitatory presynaptic neurons? If so, what are the relative population sizes? These questions should be clearly answered in the main text or the caption of Figure 11.

There are contradictory statements regarding whether inputs are shared between different neurons. Consider the sentence "To see this, let us consider a set of neuron (sic) $A$, with cardinality denoted by $|A|$, subjected to the same set of $K_e$ excitatory and $K_i$ inhibitory synchronous inputs." (lines 281-282) This implies that inputs are shared between neurons. However, presynaptic neurons are later specified by the neuron to which they send inputs. For example, $\rho_{\alpha\beta,12,kl}$ is the correlation between $k^{\text{th}}$ input of type $\alpha$ to neuron 1 and $l^{\text{th}}$ input of type $\beta$ to neuron 2. If the inputs were shared, then the $1$ and $2$ indices would be completely redundant; however, we see that $\rho_{\alpha\beta,12,kl}\neq \rho_{\alpha\beta,11,kl}\neq \rho_{\alpha\beta,22,kl}$. If populations of inputs do not overlap between the different neurons, then language suggesting they do must be removed.

In Figure 5, the color bars are somewhat misleading. Since the goal is to compare voltage variance between the six different scenarios, the six color bars should all use the same scale. Alternatively, a separate plot could be included in the figure comparing the variances between all six panels.

There are issues with the equation referencing, especially within the Supplementary Information. For example, there are several references to Eq. (45), but there are only 44 numbered equations in the manuscript.

There are many typos within the text that should be cleaned up. An important example is that the multiset defined on line 338 is called $B$, but seems to be referred to as $A_n$ in the subsequent equations. Also, in Eq. (27) the $a\subset B$ in the first square bracket should be $a\subset C$ and "time-shit" on line 312 should be "time-shift". We strongly recommend a thorough proofreading of the text to find and correct all typos.

Ultimately, the research presented in this manuscript is well-done and exciting. It provides a strong argument for the importance of synchrony in producing the correct statistics of cortical neuron activity and preemptively addresses the most important criticisms of its analytical technique. With some added clarification and careful proofreading, the manuscript will be excellent.

Reviewer #3: The paper entitled "Subthreshold moment analysis of neuronal populations driven by synchronous synaptic inputs" by Becker et al. proposes an analysis of a linear model for the evolution of the membrane potential of a neuron receiving inputs from a collection of other cells. The authors present a detailed analysis of the stationary statistics of the voltage, not limited to the first and second moments. They study, in particular, how it is affected by the presence of correlations in the inputs. This study is motivated by the observation of large fluctuations in the membrane potential in cortical intracellular recordings, a fact that apparently contradicts the large expected connectivity of those cells. Informally speaking, the presence of correlations decreases the number of independent inputs and enhances fluctuations in the potential dynamics.

I appreciate the detailed study reported in the manuscript and the care the authors brings to not only provide mathematically rigorous treatment of their models but also give interpretable formulas and insights about their results. However, there are some major points I would like to see addressed before being able to recommend publication:

1. The model studied here is simplistic. Besides the assumption of linearity, which is both a strong limitation and a necessity to be able to carry out analytics, I am worried about the absence of adequate treatment of the instability threshold. This threshold, leading to the production of an action potential, lies about 15 mV above the resting potential. If I look at the numerical plots in the paper, e.g. in Figures 5 and 8, it seems that such values are not extremely unlikely (and they should not be for a model supposed to represent a population of spiking neurons!). The authors also reports (line 396) the existence of electrophysiology experiments estimating the standard deviation of the voltage to be ~3 mV, which makes the 15mV gap not so wide. It is therefore important to take into account the reset mechanism for the voltage, which will affects its statistics. This is equivalent to a first passage-time problem, which can be (and was already) handled in the case of the linear integrate-and-fire neuron model.

2. the authors rightly emphasize the presence of strong correlations between nearby neurons in brain circuits. But they do not consider a major source for such correlations, that is, the presence of gap junctions. I think their analysis could be extended to include these interactions, as the equations remain linear, at the price of increasing the dimensionality of the tracked potential population (following the dynamics of a single neuron is not sufficient any longer).

3. my understanding of an article in PLoS Computational Biology is that detailed comparison to experiments should be included and that a purely mathematical analysis of a model is not sufficient (this is ultimately a matter of editorial decision). The authors, at different points in the manuscript, mention some measurements and statistics, such as the values of crosscorrelations, e.g. on line 518. However, experimental time traces of membrane potentials can be obtained from intracellular recordings. It would be very interesting to push further the analysis of the model through comparison with real time traces and more extended statistics that can be extracted from those data, in particular regarding time correlations (in the stationary regime). I believe this would provide further justification for the validity of the model, which, as it is now, remains rather abstract.

**Have the authors made all data and (if applicable) computational code underlying the findings in their manuscript fully available?**

Reviewer #1: Yes

Reviewer #2: None

Reviewer #3: Yes

PLOS authors have the option to publish the peer review history of their article (what does this mean?). If published, this will include your full peer review and any attached files.

Reviewer #1: No

Reviewer #2: No

Reviewer #3: No

**Figure resubmission:**
---

## [Decision Letter · Decision Letter 1]

26 Sep 2025

PCOMPBIOL-D-25-00525R1

Subthreshold moment analysis of neuronal populations driven by synchronous synaptic inputs

PLOS Computational Biology

Dear Dr. Taillefumier,

Thank you for submitting your manuscript to PLOS Computational Biology. After careful consideration, we feel that it has merit but does not fully meet PLOS Computational Biology's publication criteria as it currently stands. Therefore, we invite you to submit a revised version of the manuscript that addresses the points raised during the review process.

Please submit your revised manuscript within 30 days Nov 26 2025 11:59PM. If you will need more time than this to complete your revisions, please reply to this message or contact the journal office at ploscompbiol@plos.org. Please include the following items when submitting your revised manuscript:

We look forward to receiving your revised manuscript.

Kind regards,

Renaud Blaise Jolivet, Ph.D.

Academic Editor

PLOS Computational Biology

Marieke van Vugt

Section Editor

PLOS Computational Biology

**Additional Editor Comments:**

You will see that two of the three reviewers have recommended acceptance, but I feel that you should still address the outstanding comments from the third reviewer before I take a final decision.

**Journal Requirements:**

We have noticed that you have uploaded Supporting Information files, but you have not included a list of legends. Please add a full list of legends for your Supporting Information files after the references list.

**Reviewers' comments:**

Reviewer's Responses to Questions

**Comments to the Authors:**

Reviewer #1: The authors have thoroughly addressed my comments and those of the other reviewers.

Reviewer #2: The authors addressed all my concerns, I recommend publication.

Reviewer #3: I thank the authors for having considered my comments and criticisms. I appreciate that they have worked out the equations for coupled neurons in the presence of gap junctions, and have inserted a paragraph in the manuscript regarding this important effect. I agree that these electrical couplings, as a major effect, enhance synchrony between cells as reported in the new paragraph.

Regarding the other points:

- I am somewhat less convinced about the discussion on resetting events. In highly active areas, these events can hardly be neglected. The authors mention in their response that they have studied "the effect of including an integrate-and-fire mechanism and found that it tends to produce unrealistic skewness." I would have liked to see these results to better assess the relevance of the effects. At the very least, I would appreciated if the authors briefly include a comment about this point in the Discussion section of the final version of the manuscript.

- Lastly, about connection with experimental time traces. Comparison with real data is, to my mind, a necessity on PLoS Computational Biology articles, and would add substantial value to the present article. However, as I already wrote in my comments, this is ultimately a matter of editorial policy.

**Have the authors made all data and (if applicable) computational code underlying the findings in their manuscript fully available?**

Reviewer #1: Yes

Reviewer #2: Yes

Reviewer #3: Yes

PLOS authors have the option to publish the peer review history of their article (what does this mean?). If published, this will include your full peer review and any attached files.

Reviewer #1: No

Reviewer #2: No

Reviewer #3: No

**Figure resubmission:**
---

## [Decision Letter · Decision Letter 2]

22 Oct 2025

Dear Dr. Taillefumier,

We are pleased to inform you that your manuscript 'Subthreshold moment analysis of neuronal populations driven by synchronous synaptic inputs' has been provisionally accepted for publication in PLOS Computational Biology.

Best regards,

Renaud Blaise Jolivet, Ph.D.

Academic Editor

PLOS Computational Biology

Marieke van Vugt

Section Editor

PLOS Computational Biology

Reviewer's Responses to Questions

**Comments to the Authors:**

Reviewer #1: No further comments.

Reviewer #2: Great work!

Reviewer #3: I thank the authors for their detailed answers to my concerns about the inclusion (or lack of) of resetting effects in the model. The rebuttal letter is quite informative, and I appreciate that this point is now considered in the discussion.

Regarding the comparison to experimental data:

- I fully got from the previous version that real data were used to set some quantities in the model and some statistical comparison with theory was already present.

- What I had in mind (though this was not clearly written, and I apologise for this) is that time traces of the membrane potential dynamics are now available from experiments and I would have appreciated to see a comparison in the present paper. I went through Ref. 11 on biorxiv and I realize that this is done in this work. My intent was not to delay the publication of the present manuscript, but to improve its appeal to experimentalists (which, I am afraid, is quite limited right now).

I now recommend publication of the current manuscript.

**Have the authors made all data and (if applicable) computational code underlying the findings in their manuscript fully available?**

Reviewer #1: None

Reviewer #2: Yes

Reviewer #3: Yes

PLOS authors have the option to publish the peer review history of their article (what does this mean?). If published, this will include your full peer review and any attached files.

Reviewer #1: No

Reviewer #2: No

Reviewer #3: No

---

## [Editor Report · Acceptance letter]

PCOMPBIOL-D-25-00525R2

Subthreshold moment analysis of neuronal populations driven by synchronous synaptic inputs

Dear Dr Taillefumier,

I am pleased to inform you that your manuscript has been formally accepted for publication in PLOS Computational Biology. Your manuscript is now with our production department and you will be notified of the publication date in due course.

With kind regards,

Zsofia Freund
